# Vision as a Dialect: Unifying Visual Understanding and Generation via Text-Aligned Representations

**Jiaming Han**[12], **Hao Chen**[2†], **Yang Zhao**[2], **Hanyu Wang**[2], **Qi Zhao**[2],
**Ziyan Yang**[2], **Hao He**[12], **Xiangyu Yue**[1‡], **Lu Jiang**[2‡]

[1]CUHK MMLab  [2]ByteDance Seed

## Abstract

This paper presents a multimodal framework that attempts to unify visual understanding and generation within a shared discrete semantic representation. At its core is the Text-Aligned Tokenizer (TA-Tok), which converts images into discrete tokens using a text-aligned codebook projected from a large language model's (LLM) vocabulary. By integrating vision and text into a unified space with an expanded vocabulary, our multimodal LLM, **Tar**, enables cross-modal input and output through a shared interface, without the need for modality-specific designs. Additionally, we propose scale-adaptive encoding and decoding to balance efficiency and visual detail, along with a generative de-tokenizer to produce high-fidelity visual outputs. To address diverse decoding needs, we utilize two complementary de-tokenizers: a fast autoregressive model and a diffusion-based model. To enhance modality fusion, we investigate advanced pre-training tasks, demonstrating improvements in both visual understanding and generation. Experiments across benchmarks show that **Tar** matches or surpasses existing multimodal LLM methods, achieving faster convergence and greater training efficiency. Code, models, and data are available at https://tar.csuhan.com

## 1 Introduction

Multimodal large language models (MLLMs) [2, 12, 23, 40, 82] have demonstrated the ability of LLMs to handle visual understanding tasks within an autoregressive framework. A true MLLM is expected not only to understand images but also to generate them, laying the foundation for perception, reasoning, and interaction with the world.

For instance, MLLMs for visual understanding typically have three components: a semantic visual encoder (*e.g.*, CLIP [50]), an LLM, and a vision-to-language adapter [40, 82]. With a pre-aligned visual representation, LLaVA [40] efficiently aligns CLIP features to an LLM's latent space using just 0.6M image-text pairs and a simple linear adapter. However, visual generation representation remains an open research problem, with several key design choices outlined below.

**Separate *vs.* Shared.** Visual understanding and generation often rely on features at different levels of abstraction, leading some methods to adopt separate representations. For example, CLIP for understanding and VQVAE for generation [10, 70]. However, this separation limits unified reasoning and complicates tasks like interleaved generation or multi-turn editing. We therefore use a shared representation for both tasks, which ensure understanding and generation do not conflict but instead complement each other, as both modalities are learned within a single latent space.

**Continuous *vs.* Discrete.** Continuous visual features preserve rich information and work well for the understanding tasks but require objectives such as regression [57] or diffusion [84] for generation,

---

†Project lead. ‡Corresponding authors.

39th Conference on Neural Information Processing Systems (NeurIPS 2025).

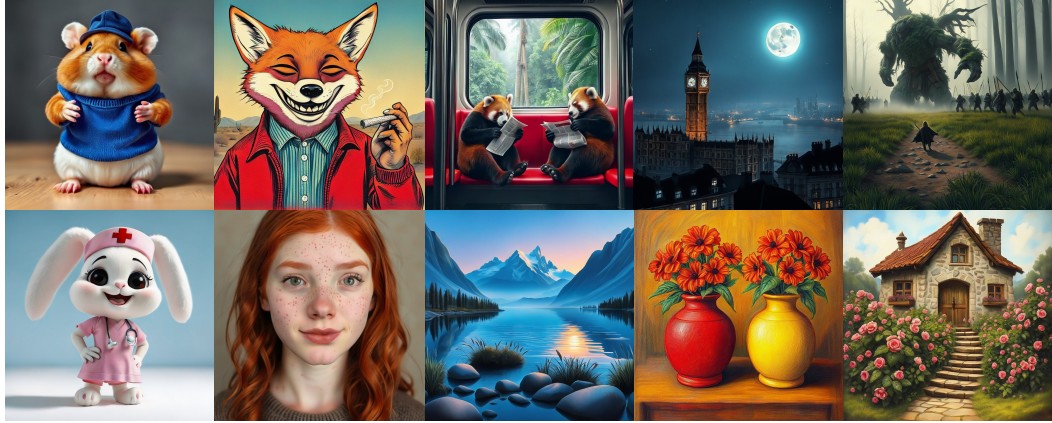

Figure 1: **Text-to-Image Generation Results**, using **Tar-7B** and a 1024 pixel de-tokenizer.

diverging from the autoregressive paradigm essential for scaling LLMs. Discrete tokens [17] align naturally with LLMs but often face quantization errors [58, 73]. We propose unifying vision and language with a shared discrete representation, simplifying the modeling paradigm, improving scalability, and reducing complexity by operating in a more efficient space. To address quantization errors, we propose a scale-adaptive representation that uses longer sequences to minimize errors and a generative de-tokenizer to enhance generation capabilities.

**Pixel *vs.* Semantic.** Pixel-level tokens (*e.g.*, VAE [17]) provide fine-grained details but are difficult to align with LLMs [58, 76, 84]. Semantic representations like CLIP [50] efficiently capture high-level semantics for interaction with LLMs but struggle to recover image details. Hybrid methods [43, 73, 83] attempt to combine both, but balancing them remains challenging. Building on the success of semantic representations in visual understanding, we extend their use to visual generation, enabling faster convergence and simplifying the unification of understanding and generation.

This paper studies **Text-aligned representation (Tar)**, a *fully discrete and semantic* representation that attempts to unify visual understanding and generation within a *shared* space. Central to our method is the **Text-Aligned Tokenizer (TA-Tok)**, which converts images into discrete tokens using a text-aligned codebook initialized from an LLM's vocabulary and adapted to vision through learnable projection layers. This approach enables seamless cross-modal input and output without relying on modality-specific designs, and supports advanced multimodal reasoning within a unified framework.

To balance efficiency and detail, we introduce **Scale-Adaptive Pooling and Decoding**, which lets the model adjust token length as needed: coarse-grained tokens for efficient generation and fine-grained tokens for detailed understanding. For decoding, we use two complementary **Generative De-Tokenizers**: a fast autoregressive (AR) model for discrete VAE latents, and a diffusion-based model for continuous VAE latents. The AR de-tokenizer is fast and works well with discrete LLM tokens, while the diffusion de-tokenizer leverages powerful pretrained image generators [75] for high-quality outputs. Together, they provide a flexible balance between speed, compatibility, and visual fidelity. Besides the common understanding (image-to-text) and generation (text-to-image) tasks, we further improve modality fusion with new pre-training tasks like image-to-image, text-image-to-image, boosting both visual understanding and generation.

We summarize our key features as follows:

- We propose a Text-Aligned Tokenizer that unifies visual understanding and generation in a shared semantic, discrete space. This multimodal framework eliminates the need for modality-specific designs and allows seamless input and output across modalities through a common interface.

- Our Scale-Adaptive Pooling and Decoding provide flexible control over visual detail for different tasks. Additionally, we introduce Generative De-Tokenizers that generate images from discrete semantic tokens using either autoregressive or diffusion-based models.

- We explore advanced pre-training schemes to enable both visual understanding and generation within a single model, achieving strong performance on various benchmarks.

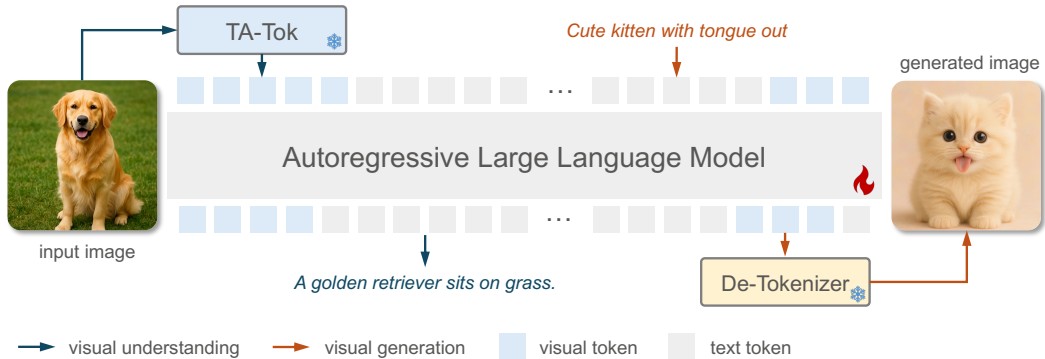

Figure 2: **Architecture of Tar,** a multimodal LLM that unifies visual understanding and generation in an autoregressive paradigm. Refer to Sec. 3.3 for training and inference detail.

## 2 Related Work

**Unified Multimodal Large Language Models.** With the development of LLMs [1, 4, 46, 63], MLLMs have attracted a lot of research interest due to their strong multimodal understanding and reasoning capabilities [2, 12, 30, 40, 82]. Beyond visual understanding, several recent works [14, 20, 57, 67, 70, 76, 84] attempt to integrate both visual understanding and generation within a unified MLLM. Emu2 [57] enables LLMs to generate CLIP embeddings, which are decoded into images using a diffusion model. Emu3 [69] and Chameleon [58] use VQVAE [17] as both the visual encoder and decoder, allowing unified next token prediction across images and text. However, VQVAE's focus on pixel dependency limits MLLM's ability to handle both low-level image details and high-level semantics. Show-o [76] and Transfusion [84] integrate diffusion objectives into LLMs for image generation, but this design breaks the autoregressive paradigm and complicates the unification of the two tasks. Janus [10, 70] takes a modular approach with separate encoders for understanding and generation, but results in distinct modalities for image understanding and generation, which can hinder tasks like multi-turn image editing and interleaved generation. VILA-U [73] and UniTok [43] train a fused tokenizer using both pixel reconstruction and image-text alignment losses, but the model struggles to converge optimally for both tasks. ILLUME [67] applies vector quantization to a semantic visual encoder, using discrete tokens for image generation. However, ILLUME still relies on continuous visual features for visual understanding, resulting in separate encoders for the two tasks. In contrast, we propose a fully discrete, semantic and shared representation that unifies understanding and generation within a single MLLM.

**Visual Tokenization.** For visual generation, both continuous and and discrete VAEs [26, 65] are popular visual tokenizers. The continuous VAE [26] is used as the default tokenizer for diffusion models [27, 52], while discrete VAE [65] is used for masked generative models [6] and autoregressive models [17, 61, 68]. For visual understanding, the best practice is employing a continuous, semantic tokenizer like CLIP [50], SigLIP [81] and DINO [5]. However, it is still unclear what type of tokenizer is good for both visual understanding and generation. As discussed in Sec. 1, some works [58, 69] leverage continuous tokenizers [57, 84], while others use discrete tokenizers [58, 69, 76]. Some works use pixel-level tokenizers [58, 69], other works leverage semantic tokenizers [57, 67] or hybrid tokenizers [43, 73]. In contrast, we leverage a discrete and LLM-aligned tokenizer for visual understanding and generation. The generated visual tokens are then decoded with autoregressive or diffusion de-tokenizer for high quality image generation.

## 3 Method

In this section, we first propose **Text-Aligned Tokenizer (TA-Tok)**, which converts images into text-aligned, scale-adaptive, discrete tokens in Sec. 3.1. In Sec. 3.2, we introduce **Generative De-Tokenizers**, leveraging powerful generative models (*e.g.*, autoregressive [56] and diffusion models [74]) for decoding high-quality images from our text-aligned visual conditions. Built on TA-Tok and de-tokenizers, we design a unified MLLM with the simple next token prediction approach.

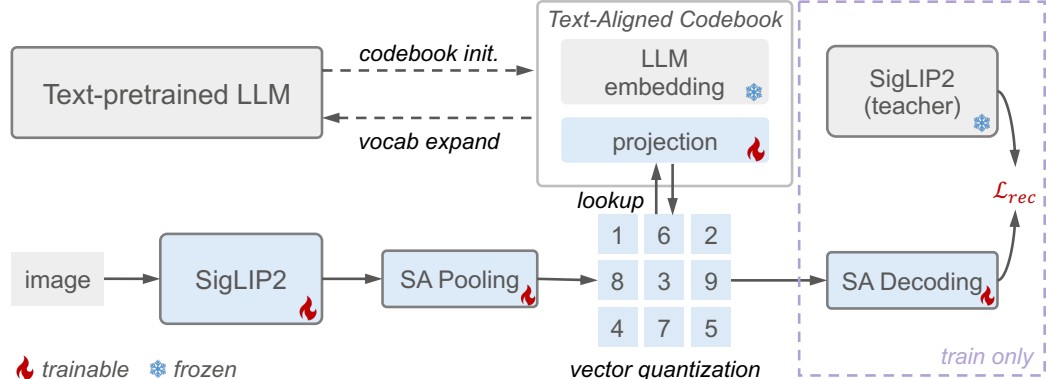

Figure 3: **Text-Aligned Tokenizer.** An input image is first encoded into continuous tokens using a SigLIP2 vision model [64], followed by Scale-Adaptive (SA) Pooling to adjust spatial resolution. These tokens are then discretized via a Text-Aligned Codebook, which is initialized from LLM embeddings. To guide training, SA Decoding reconstructs the pre-quantized tokens and is supervised by a SigLIP2 teacher model using a reconstruction loss $\mathcal{L}_{rec}$. The decoder and teacher are only used during training and discarded at inference. Once trained, the codebook can serve as a visual vocabulary of the LLM.

The overall architecture is shown in Fig. 2. Finally, we illustrate in Sec. 3.4 the training recipe, especially the proposed unified pretraining tasks.

## 3.1 Text-Aligned Tokenizer

TA-Tok is designed to align visual representations to LLM's latent space. Below we will introduce the basic concept and architecture of TA-Tok. The overall architecture of TA-Tok is shown in Fig. 3.

**Vector Quantization (VQ)** is a technique to discretize continuous representations into a finite set of tokens, thereby transforming high-dimensional vectors into a set of quantized representations. Given a continuous input vector $\mathbf{z}_I = \mathcal{E}(I)$, which is obtained from an image $I$ encoded by a visual encoder $\mathcal{E}$, the goal is to map it to the closest vector in a codebook $\mathcal{C}$. The quantization process is formulated as: $\mathbf{z}_q = \mathrm{argmin}_{\mathbf{c} \in \mathcal{C}} \|\mathbf{z}_I - \mathbf{c}\|^2$, where $\mathbf{C} = \{\mathbf{c}_1, \mathbf{c}_2, \ldots, \mathbf{c}_K\}$ and $K$ is the number of codebook entries. The aim is to map the input $\mathbf{z}$ to the most representative vector $\mathbf{c}_k$ in $\mathbf{C}$.

**Text-Aligned Codebook.** Traditional VQ codebooks are usually random initialized. To align visual and textual tokens in the latent space of LLM, we initialize the VQ codebook using the token embeddings of a pretrained LLM $\mathbf{E} \in \mathbb{R}^{K \times D} = \{\mathbf{e}_1, \mathbf{e}_2, \ldots, \mathbf{e}_K\}$ and a projection matrix $\mathbf{W} \in \mathbb{W}^{D \times D} = \{\mathbf{w}_1, \mathbf{w}_2 \ldots, \mathbf{w}_D\}$. The codebook $\mathbf{C} \in \mathbb{R}^{K \times D}$ is defined as:

$$\mathbf{C} = \mathbf{E}\mathbf{W} = \{\mathbf{E}\mathbf{w}_1, \mathbf{E}\mathbf{w}_2, \ldots, \mathbf{E}\mathbf{w}_D\}, \tag{1}$$

$$\mathbf{E}\mathbf{w}_d = \{\mathbf{e}_1\mathbf{w}_d, \mathbf{e}_2\mathbf{w}_d, \ldots, \mathbf{e}_K\mathbf{w}_d\}. \tag{2}$$

Note the LLM embeddings $\mathbf{E}$ are always frozen and we only train the projection matrix $\mathbf{W}$, which ensures each codebook entry is a projected version of a corresponding LLM token embedding. By grounding the visual codebook in the LLM's latent space, this design ensures that visual tokens are semantically aligned with textual tokens, facilitating a unified representation across modalities.

Since LLM vocabularies are often large (*e.g.*, 150K for Qwen [78]), using the entire embedding set as codebook is computational impractical. We select the top-$k$ most representative embeddings based on their average distance to others, ensuring broad semantic coverage with minimal redundancy:

$$\mathbf{E} = \mathrm{argsort}\left(\frac{1}{N}\sum_{j=1}^{N}\frac{\mathbf{e}_i \cdot \mathbf{e}_j}{\|\mathbf{e}_i\|\|\mathbf{e}_j\|}\right)[:k], \tag{3}$$

where $N$ is the vocabulary size of the LLM and $i, j \in \{1, 2, \ldots, N\}$.

**Scale-Adaptive Pooling and Decoding.** Different tasks vary in their need for visual details, *e.g.*, understanding and editing often rely on fine-grained features, while generation may benefit from

coarser representations [18]. To accommodate these differences, we introduce Scale-Adaptive Pooling (SAP) and Decoding (SAD) to extract multi-granularity features. Given image features $\mathbf{z}_I$, we apply SAP with scale factor $s \in \{1, 2, 3\}$ to obtain $\mathbf{z}_I^p = \text{SAP}(\mathbf{z}_I, s)$, allowing control over visual details based on task needs or compute budget. During decoding, we follow SigLIP2 [64] by resizing 2D positional embedding to match the input scale, enabling the ViT decoder [64] to process multi-scale latent features effectively.

**Architecture and Training Objective.** As shown in Fig. 3, TA-Tok consists of a SigLIP2 encoder, a SAP, a Text-Aligned Codebook, a SAD and a SigLIP2 teacher model. SAP is implemented via adaptive pooling, while SAD uses a lightweight ViT decoder with three ViT blocks. TA-Tok is trained with a combination of feature reconstruction loss $\mathcal{L}_{rec}$ and codebook losses $\mathcal{L}_{code}$. The reconstruction loss encourages semantic alignment between the decoded features and the SigLIP2 teacher output, defined as: $\mathcal{L}_{rec} = 1 - \frac{\mathbf{z}_y \cdot \hat{\mathbf{z}}_y}{\|\mathbf{z}_y\| \|\hat{\mathbf{z}}_y\|}$, where $\mathbf{z}_y$ and $\hat{\mathbf{z}}_y$ are features from SAD and SigLIP2 teacher, respectively. The codebook losses $\mathcal{L}_{code}$ ensures the quantized features are close to the codebook entries. We freeze the LLM token embeddings $\mathbf{E}$ and only train the projection matrix $\mathbf{W}$. The loss is defined as:

$$\mathcal{L}_{code} = \|\text{sg}(\mathbf{C}) - \mathbf{z}_q\|_2^2 + \|\mathbf{C} - \text{sg}(\mathbf{z}_q)\|_2^2 = \|\mathbf{E} \cdot \text{sg}(\mathbf{W}) - \mathbf{z}_q\|_2^2 + \|\mathbf{E}\mathbf{W} - \text{sg}(\mathbf{z}_q)\|_2^2, \quad (4)$$

where $\text{sg}(\cdot)$ denotes the stop-gradient operation. During training, the SigLIP2 encoder, SAP and SAD are jointly optimized, while the SigLIP2 teacher remains frozen.

## 3.2 Generative De-Tokenizer

Since TA-Tok produces only semantic tokens without image generation capability, we introduce a Generative De-Tokenizer to decode high-quality images from its quantized outputs. As shown in Fig. 4, we propose two variants: autoregressive de-tokenizer (AR-DTok) and diffusion de-tokenizer (Dif-DTok), corresponding to two dominant paradigms in image generation.

**Autoregressive De-Tokenizer.** In Fig. 4 (a), we formulate image decoding as an autoregressive generation task. Let $\theta_{AR}$ be the parameters of AR-DTok, and $\mathbf{y} = [y_1, y_2, \ldots, y_T]$ be the image tokens (in yellow) from a VQVAE encoder. The AR-DTok predicts each token $y_t$ conditioned on the semantic visual tokens $\mathbf{z}_q$ (in blue) from TA-Tok and previous generated tokens $y_{<t}$:

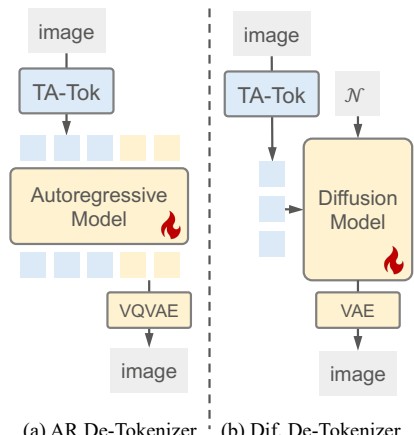

$$\mathcal{L}(\theta_{AR}) = -\sum_{t=1}^{T} \log p(y_t | \mathbf{z}_q, y_{<t}; \theta_{AR}). \quad (5)$$

(a) AR De-Tokenizer    (b) Dif. De-Tokenizer

Figure 4: **Architecture of Generative De-Tokenizer Variants.**

**Diffusion De-Tokenizer.** Fig. 4 (b) illustrates the second variant, where the quantized tokens $\mathbf{z}_q$ serve as conditioning input to a diffusion model via cross attention, similar to text conditioning in traditional diffusion models [52, 54]. Let $F$ be the diffusion model parameterized by $\theta_{dif}$, which denoised a noised latent $y_t$ to predict the original clean latent $y_0$. The training objective is:

$$\mathcal{L}(\theta_{dif}) = \mathbb{E}_t[\|F(y_t, \mathbf{z}_q; \theta_{dif})\|^2]. \quad (6)$$

In practice, we reuse most parameters of a pretrained diffusion model, simply replacing the original text condition with TA-Tok's visual tokens, enabling high-fidelity image synthesis with minimal adaption.

**Discussion.** AR-DTok and Dif-DTok offer complementary strengths. AR-DTok aligns naturally with our discrete condition $\mathbf{z}_q$, enabling a unified autoregressive modeling framework. It also benefits from faster inference due to its sequential decoding. In contrast, Dif-DTok leverages powerful pretrained diffusion models, allowing quick adaptation to new conditions like $\mathbf{z}_q$. While it is more computational intensive at inference time, it requires less training data and remains competitive with AR-DTok in overall performance. Its strong generation prior also makes it particularly useful in tasks involving complex scenes or where high visual fidelity is desired.

Table 1: **Results on Visual Understanding Benchmarks,** including POPE [33], MME [19], MMB [41], SEED [28], GQA [25] and MMMU [79]. Token: Token type, including Continuous (C), Discrete (D), Semantic (S), Pixel (P) and Hybrid (H).

| Model | # LLM | Token | POPE↑ | MME-P↑ | MME-C↑ | MMB↑ | SEED↑ | GQA↑ | MMMU↑ |
|---|---|---|---|---|---|---|---|---|---|
| *Understanding Only Model* | | | | | | | | | |
| LLaVA-Phi [76] | 1.3B | C,S | 84.1 | 1128 | - | - | - | 56.5 | 30.7 |
| MobileVLM-V2 [11] | 1.4B | C,S | 84.3 | 1303 | - | 57.7 | - | 59.3 | - |
| DeepSeekVL [42] | 1.3B | C,S | 88.3 | 1307 | - | 64.6 | - | 59.3 | 33.8 |
| MiniGemini [32] | 2B | C,S | 83.9 | 1341 | - | 59.8 | - | 59.9 | - |
| LLaVA-v1.5 [38] | 7B | C,S | 85.9 | 1511 | - | 64.3 | 58.6 | 62.0 | 35.4 |
| Qwen-VL-Chat [2] | 7B | C,S | - | 1488 | - | 60.6 | 58.2 | 57.5 | - |
| Emu3-Chat [69] | 8B | D,P | 85.2 | 1244 | - | 58.5 | 68.2 | 60.3 | 31.6 |
| *Unified Model* | | | | | | | | | |
| Show-o [76] | 1.3B | D,P | 80.0 | 1097 | 248 | - | - | 58.0 | 26.7 |
| Harmon [72] | 1.5B | C,H | 87.6 | 1155 | 321 | 65.5 | 67.1 | 58.9 | **38.9** |
| Janus [70] | 1.5B | C,S | 87.0 | 1338 | 222 | 69.4 | 63.7 | 59.1 | 30.5 |
| Janus-Pro [10] | 1.5B | C,S | 86.2 | **1444** | 268 | **75.5** | 68.3 | 59.3 | 36.3 |
| D-Dit [35] | 2.0B | C,P | 84.0 | 1125 | - | - | - | 59.2 | - |
| **Tar (Ours)** | 1.5B | D,S | **88.4** | 1390 | **342** | 65.6 | **70.4** | **61.1** | 36.0 |
| ILLUME [67] | 7B | C,S | **88.5** | 1445 | - | 65.1 | 72.9 | - | 38.2 |
| Chameleon [58] | 7B | D,P | - | - | - | - | - | - | 22.4 |
| LWM [37] | 7B | D,P | 75.2 | - | - | - | - | 44.8 | - |
| Liquid[71] | 7B | D,P | 81.1 | 1119 | - | - | - | 58.4 | - |
| UniTok [43] | 7B | D,H | 83.2 | 1448 | - | - | | 61.1 | - |
| VILA-U [73] | 7B | D,H | 85.8 | 1402 | - | - | 59.0 | 60.8 | - |
| Janus-Pro [10] | 7B | C,S | 87.4 | 1567 | 260 | **79.2** | 72.1 | **62.0** | 41.0 |
| MetaMorph [62] | 8B | C,S | - | - | - | 75.2 | 71.8 | - | **41.8** |
| **Tar (Ours)** | 7B | D,S | 87.8 | **1571** | **355** | 74.4 | **73.0** | 61.3 | 39.0 |

## 3.3 Unified Multimodal Modeling

Built on TA-Tok and Generative De-Tokenizers, we propose a unified MLLM, **Tar (Text-aligned representation)**, with a simple autoregressive objective and eliminating the need for modality-specific designs. The architecture is shown in Fig. 2.

**Visual Embedding Initialization.** We represent both text and images as discrete tokens in a shared vocabulary by expanding the LLM's text embedding matrix $\mathbf{E} \in \mathbb{R}^{M \times D}$ with a visual token set $\mathbf{C} \in \mathbb{R}^{K \times D}$. Rather than randomly initializing $\mathbf{C}$, we use our Text-Aligned Codebook $\mathbf{C} = \mathbf{EW}$ (with $\mathbf{W} \in \mathbb{W}^{D \times D}$) as the visual embeddings. Since $\mathbf{C}$ and $\mathbf{E}$ share the same dimensionality, we can simply set: $\{\mathbf{E}, \mathbf{C}\} = \{\mathbf{E}, \mathbf{EW}\}$. This operation eliminates any extra embedding alignment stage. The unified embedding enables the LLM to natively process and generate both modalities without additional connectors or decoding heads.

**Training.** We train Tar with the standard Cross-Entropy loss over a mixed sequence of text and visual tokens, Given a target sequence $\mathbf{u} = [u_1, u_2, \ldots, u_N]$ ($u_i$ may be a text or visual token) and model parameter $\theta$, the loss is: $\mathcal{L}_{CE} = -\sum_{i=1}^{N} \log(u_i|\mathbf{u}_{<i}; \theta)$.

**Inference.** At inference time, Tar can either take tokens from both modalities as input or generate them. **(a)** Visual understanding: Feed TA-Tok's visual tokens $\mathbf{z}_q$ and any text prompt into LLM, and generate text tokens for image captioning, visual question answering, *etc*. **(b)** Visual generation: Provide a text prompt and autoregressively sample a sequence of visual tokens, then pass these tokens to a de-tokenizer to decode the final image.

## 3.4 Training Recipe

**Data Curation.** Our training data consists of image, text and multimodal datasets. Since open-sourced datasets for image, text and image-to-text tasks are widely available [13, 53, 60], our focus is on curating high-quality data for image generation. The pipeline includes: **(1) Image caption.** We use Qwen2.5-VL [3] to generate rich, detailed captions for general image datasets [13, 15, 51]. **(2) Synthetic image generation.** We adopt FLUX [27] to generate high quality images based on real user prompts [55, 66] and image captions from Step 1, which yield diverse, prompt-aligned content. In total, we curate a dataset of 23M high-quality text-image pairs for training.

Table 2: **Results on Visual Generation Benchmarks**. *: We use an AR-DTok with 256px resolution. Due to space limit, we omit some metrics and put full results to Appendix Sec. F.

| Method | GenEval [22] | | | | DPG Bench [24] | | | |
|---|---|---|---|---|---|---|---|---|
| | Two Obj. | Counting | Color Attri. | Overall↑ | Entity | Attribute | Relation | Overall↑ |
| *Generation Only Model* | | | | | | | | |
| Emu3-Gen [69] | 0.71 | 0.34 | 0.21 | 0.54 | 86.68 | 86.84 | 90.22 | 80.60 |
| SDXL [48] | 0.74 | 0.39 | 0.23 | 0.55 | 82.43 | 80.91 | 86.76 | 74.65 |
| Playground v2.5 [29] | - | - | - | - | 82.59 | 81.20 | 84.08 | 75.47 |
| Hunyuan DiT [34] | - | - | - | - | 80.59 | 88.01 | 74.36 | 78.87 |
| PixArt-Σ [8] | - | - | - | - | 82.89 | 88.94 | 86.59 | 80.54 |
| DALLE3 [36] | 0.87 | 0.47 | 0.45 | 0.67 | 89.61 | 88.39 | 90.58 | 83.50 |
| SD3-Medium [16] | 0.94 | 0.72 | 0.60 | 0.74 | 91.01 | 88.83 | 80.70 | 84.08 |
| SANA-1.5 [75] | 0.93 | 0.86 | 0.65 | 0.81 | - | - | - | **84.70** |
| *Unified Model* | | | | | | | | |
| Chameleon-7B [58] | - | - | - | 0.39 | - | - | - | - |
| LWM-7B [37] | 0.41 | 0.46 | 0.15 | 0.47 | - | - | - | - |
| SEED-X-13B [21] | 0.58 | 0.26 | 0.14 | 0.49 | - | - | - | - |
| Show-o-1.3B [76] | 0.52 | 0.49 | 0.28 | 0.53 | - | - | - | - |
| Transfusion-7B [84] | - | - | - | 0.63 | - | - | - | - |
| D-DiT-2B [35] | 0.80 | 0.54 | 0.50 | 0.65 | - | - | - | - |
| ILLUME-7B [67] | 0.86 | 0.45 | 0.28 | 0.61 | - | - | - | - |
| Janus-1.3B [70] | 0.68 | 0.30 | 0.42 | 0.61 | 87.38 | 87.70 | 85.46 | 79.68 |
| Janus-Pro-1B [10] | 0.82 | 0.51 | 0.56 | 0.73 | 88.63 | 88.17 | 88.98 | 82.63 |
| Harmon-1.5B [72] | 0.86 | 0.57 | 0.48 | 0.76 | - | - | - | - |
| Janus-Pro-7B [10] | 0.89 | 0.59 | 0.66 | 0.80 | 88.90 | 89.40 | 89.32 | 84.19 |
| **Tar-1.5B* (Ours)** | 0.91 | 0.76 | 0.51 | 0.76 | 89.35 | 86.91 | 93.50 | 82.96 |
| *w/ Self Reflect* | 0.92 | 0.77 | 0.55 | 0.78 | 88.48 | 87.83 | 93.38 | 84.10 |
| **Tar-7B* (Ours)** | 0.92 | 0.83 | 0.65 | 0.84 | 88.62 | 88.05 | 93.98 | 84.19 |
| *w/ Self Reflect* | 0.93 | 0.86 | 0.70 | **0.85** | 88.60 | 88.78 | 93.59 | 84.65 |

**Tokenizer and De-Tokenizer Training.** TA-Tok is trained on 100M raw and 100M aesthetic-filtered images from LAION [53], balancing its capability in visual understanding and generation. For AR-DTok, due to the lack of pretrained models, we train from scratch at 256px resolution and finetune to 512px and 1024px using 50M aesthetic images from LAION [53] and 23M synthetic images. For Dif-DTok, we initialize from pretrained SANA-0.6B [74], allowing direct finetuning at 512px resolution on the 23M synthetic dataset.

**Unified MLLM Pretraining.** Our LLM is trained on a diverse mix of data types, including standard image-to-text (I→T), text-to-image (T→I) and text-only (T→T) tasks. To further bridge the gap between visual understanding and generation, we introduce two additional task type: text-image-to-image (TI→I) and image-to-image (I→I). For I→I, we use FLUX to generate two images from the same prompt. For TI→I, we let Qwen [78] to split a prompt into a text with an image placeholder and a reference caption. For example, *"A dog running on the grass"* becomes: *"A dog running on <image>"* and *"the grass"*. The goal is to generate an image based on the input image (I→I) or image and text (TI→I), which encourage deeper multimodal integration and empirically, we observe that they accelerate convergence and improve alignment between understanding and generation.

**Supervised MLLM Finetuning.** For visual understanding, we use open-source instruction tuning datasets from LLaVA-v1.5 [38] and LLaVA-Next [39]. For visual generation, we filter a high-quality subset from our pretraining datasets with CLIP score >0.25. Additional, we collect a few human-preferred and task-aligned examples using advanced generation models [49, 75]. For more details on the training datasets, please refer to Appendix Sec. C.

# 4 Experiments

## 4.1 Experiment Details

For TA-Tok, we use `siglip2-so400m-patch14-384` [64] as the visual encoder and a three-layers ViT [64] as the decoder. We select 65536 tokens from Qwen2.5 [78] as LLM embeddings in TA Codebook. An 384×384 image is encoded into $\{729, 169, 81\}$ tokens at scale $\{1, 2, 3\}$. For AR-DTok, we adopt the LLaMA architecture [63] implemented in Llamagen [56]. The autoregressive model is trained from scratch. The discrete VAE for image decoding is pretrained by Llamagen. For Dif-DTok, we use pretrained SANA-0.6B [74] and only finetune the cross attention and condition

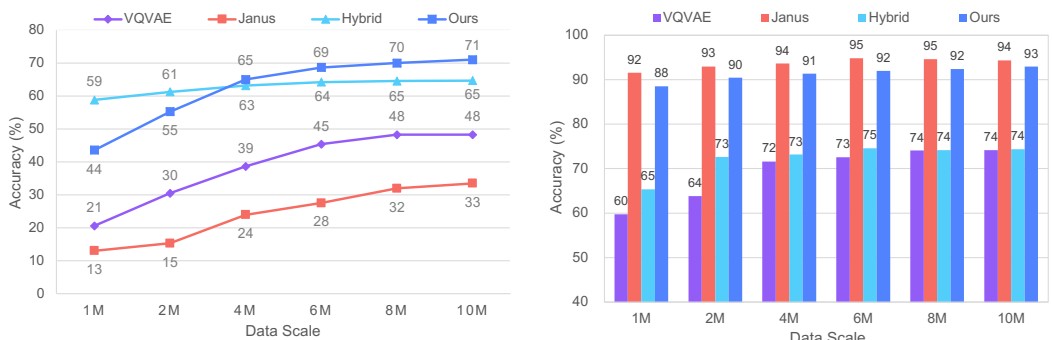

Figure 5: **Comparisons of Visual Representations on Generation and Understanding Tasks.**
**Left**: Generation performance evaluated by DPG Score [24]. **Right**: Understanding performance measured by the harmonic mean over benchmarks [19, 25, 28, 33].

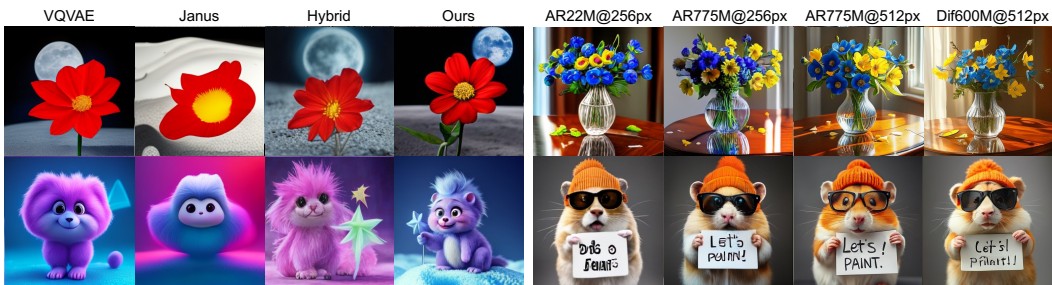

Figure 6: **Qualitative Comparison of Different Representations (Left) and De-Tokenizers (Right).**

embedding layers. For MLLM, we adopt Qwen2.5-Instruct [78] as the backbone LLM. The LLM is fully finetuned at both pretraining and finetuning stages. For training, we randomly select a scale from $\{1, 2, 3\}$. For inference, we set it to 1 if not specified. See Appendix Sec. D for more detail.

## 4.2 Main Results

**Visual Understanding.** As shown in Tab. 1, **Tar** model demonstrates strong visual understanding performance across a broad range of benchmarks. Our 1.5B model surpasses most understanding only models and unified models at 1.5B/7B scale. Our 7B model matches the performance of Janus-Pro-7B, a state-of-the-art model with continuous visual tokens. These results confirm that a unified modeling framework using fully discrete tokens, when coupled with strong text-aligned representation, can match and even surpass specialized continuous-token models in visual understanding.

**Visual Generation.** In Tab. 2, **Tar** achieves strong performance on both GenEval [22] and DPG Bench [24]. On GenEval, it reaches 0.76/0.84 overall score, surpassing all unified models. On DPG Bench, Tar-1.5B achieves 82.96 score, outperforming Janus-Pro-1B and even approaching Janus-Pro-7B. To fully leverage Tar's multimodal reasoning ability, we propose a **Self Reflect** strategy, which allows the model to assess image-prompt alignment using its own visual understanding capabilities, leading to further performance improvement. The image generation results are visualized in Fig. 1 and Appendix Fig. 8. For more details about the Self Reflect strategy, please refer to Appendix Sec. E.

## 4.3 Comparisons with Other Visual Tokenization Methods

In this section, we compare our text-aligned representation (TA-Tok) with other visual tokenization methods for unified visual understanding and generation. We considering the following recent methods [43, 69, 70]: **(a) VQVAE**: A full pixel-level representation. We use a pretrained VQVAE from Llamagen [56], which transforms images into discrete tokens for multimodal modeling. **(b) Janus**: Use separate encoders for understanding (SigLIP2) and generation (VQVAE). Note our implementation follows this approach but differs from the original one [70]. **(c) Hybrid**: A hybrid model that maintains both pixel and semantic representations. We follow UniTok [43] to train a

Table 3: **MLLM Embedding Initialization.** Und: Harmonic mean of understanding benchmarks [19, 25, 28, 33]. Gen: DPG score [24].

| emb. init | Data | Und | Gen |
|---|---|---|---|
| random | 25M | 91.1 | 69.6 |
| pre-align | 50M+25M | 89.8 | 70.3 |
| TA-Tok | 25M | **92.9** | **70.6** |
| pre-align | 100M+25M | 91.5 | 72.5 |

Table 4: **Ablation of Different De-Tokenizers.**

| Type | Size | Res | GenEval | DPG |
|---|---|---|---|---|
| | 775M | 256 | 0.76 | **83.0** |
| | 775M | 512 | **0.79** | 82.7 |
| AR | 775M | 1024 | **0.79** | 82.6 |
| | 111M | 256 | 0.75 | 82.2 |
| | 22M | 256 | 0.74 | 81.7 |
| Dif. | 600M | 512 | 0.77 | 82.4 |

Table 5: **The Effect of Scale-Adaptive Pooling.** Und: Results on understanding tasks with 10M data model. Right: Different data scale on DPG Bench. *: Model after supervised finetuning.

| #Token | Und | Data Scale on Generation | | | |
|---|---|---|---|---|---|
| | Avg | 10M | 20M | 40M* | Avg |
| 729 | **92.9** | 70.6 | 73.8 | **82.3** | 75.6 |
| 169 | 89.1 | 72.7 | 74.0 | 82.2 | **76.3** |
| 81 | 86.6 | **73.0** | 73.5 | 80.1 | 75.5 |

Table 6: **Separate Train *vs*. Joint Train.** We evaluate the performance of different visual representations under joint understanding & generation training or separate training.

| Model | Und Δ (sep. → joint) | Gen Δ (sep. → joint) |
|---|---|---|
| Janus | -0.6 (**95.0** → 94.4) | -0.1 (**33.5** → 33.4) |
| VQVAE | +0.3 (73.9 → **74.2**) | +8.1 (40.1 → **48.2**) |
| Ours | +0.1 (92.8 → **92.9**) | +5.3 (65.3 → **70.6**) |

hybrid tokenizer on 100M image-text pairs [53], using both pixel reconstruction loss and image-text alignment loss. For training MLLMs with these representations, we sample a subset of our training data for controlled experiments: 10M T2I data, 10M I2T data and 5M text-only data. All models are trained with the same configuration and tested on visual understanding [19, 25, 28, 33] and generation tasks [24].

**TA-Tok Outperforms in Visual Generation.** As shown in the left of Fig. 5, TA-Tok excels in visual generation. It achieves the highest performance across all data scales, with VQVAE, Janus and TA-Tok showing similar convergence curves. Although Hybrid starts with higher performance, it does not scale effectively with increasing data. Notably, Janus underperforms VQVAE, likely due to conflicts between the understanding and generation tasks. Besides, we found TA-Tok generates high-fidelity images (see appendix), while models using pixel representations are struggle with image details, suggesting semantic representation is more suitable for LLM-based image generation.

**TA-Tok Matches Continuous-Semantic Representations in Visual Understanding.** In visual understanding tasks (right of Fig. 5), Janus performs slightly better due to its continuous semantic encoder. However, TA-Tok is close behind with scores of 93 *vs*. 94 at 10M data. Hybrid, despite joint training with pixel and semantic losses, performs similar to VQVAE, not achieving the expected performance due to its bias to pixel representation. Overall, TA-Tok achieves the best balance between visual generation and understanding tasks, outperforms other methods in both domains.

## 4.4 Ablation Experiments

In this section, we conduct ablation experiments on our key designs and demonstrate the effectiveness of the proposed method. We use the same subset of training data in Sec. 4.3 if not specified.

**Text-Aligned Codebook for Better MLLM Initialization.** One benefit of our TA-Tok is the learned codebook can be directly transferred to LLM for image embedding initialization, as discussed in Sec. 3.3. In Tab. 3, we show that initializing the LLM embeddings with TA Codebook leads to better performance on both understanding and generation tasks compared to random initialization. Another common approach pre-align, used in previous works [10, 70], involves a separate alignment stage with additional training data. While pre-align with 50M extra data can match TA-Tok's performance on generation, which further highlights the efficiency and effectiveness of our approach.

**Generative De-Tokenizer.** Tab. 4 shows that AR-DTok performs well across resolution and scales, with 512px offering the best trade-off between quality and efficiency. Larger models (775M) perform slightly better, though smaller variants remain competitive. Dif-DTok achieves similar scores at 512px, but adapts quickly to generating high-fidelity images thanks to its pretrained diffusion backbone.

However, Fig. 6 (right) shows that the visual quality of generated images varies between de-tokenizers, even when their scores are similar.

**Scale-Adaptive Pooling for Multi-Granularity Visual Tasks.** The design of SAP makes our model produce multi-granularity visual tokens. In Tab. 5, we show that visual understanding tasks require more tokens to capture image details. However, adding more visual tokens does not significantly improve image generation performance. On the contrary, longer visual sequences make it harder for LLMs to learn (*e.g.*, 81 tokens are optimal at 10M training data). While longer sequences can enhance T2I performance as the training data increase, we found 169 tokens are already sufficient, which is similar to conventional T2I models [16, 75].

**Shared Representation Unifies Visual Understanding and Generation.** Previous works [62, 72] studied the mutual effect of both tasks. In Tab. 6, we further show that the two tasks can benefit each other as long as we use a shared visual representation. For Janus-style representation, jointly training does not decrease the performance of each task under separate training. But the two tasks can benefit each other with a shared representation such as VQVAE and our text-aligned representation, around 8.1% and 5.3% improvement on the generation task, respectively. We also notice an imbalance in the improvements between the understanding and generation tasks, which may be attributed to two factors: **(a)** Task and supervision mismatch. Generation tasks benefit more from joint train because they requires understanding before generating coherent images. Understanding tasks, on the other hand, are more discriminative and do not directly benefit from generation supervision. **(b)** Task difficulty and data requirements. Understanding tasks require less data to achieve strong performance, whereas generation tasks are more complex and require more data. Thus, joint training benefits generation tasks more significantly.

**Unified Multimodal Pretraining with Advanced Tasks.** In the right table, we demonstrate that the proposed TI2I and I2I tasks can further improve the generation performance, thus narrowing the gap between understanding and generation. Ratio: the data ratio of T2I *vs.* I2I *vs.* TI2I.

| Task | Ratio | Und | Gen |
|------|-------|-----|-----|
| baseline | (3:0:0) | 89.2 | 66.4 |
| I2I | (2:1:0) | 89.4 | 68.5 |
| TI2I | (2:0:1) | 89.8 | 70.1 |
| **I2I+TI2I** | (4:1:1) | **89.8** | **70.7** |

## 5    Conclusion

We introduced Tar, a unified model that bridges visual understanding and generation using a shared, discrete, text-aligned representation. By aligning image tokens with LLM embeddings via TA-Tok, and supporting scale-adaptive tokenization and generative de-tokenizers, Tar achieves strong performance across both tasks without modality-specific designs. Our results show that a fully discrete, semantic representation enables efficient and effective multimodal learning, moving toward true unification of vision and language.

## Acknowledgments

This work is partially supported by the National Natural Science Foundation of China (Grant No. 62306261), and the SHIAE Grant (No. 8115074). This study was supported in part by the Centre for Perceptual and Interactive Intelligence, a CUHK-led InnoCentre under the InnoHK initiative of the Innovation and Technology Commission of the Hong Kong Special Administrative Region Government. This work is also partially supported by Hong Kong RGC Strategic Topics Grant STG1/E-403/24-N and CUHK-CUHK(SZ)-GDST Joint Collaboration Fund YSP26-4760949.

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

# Appendix

## A  Overview

## B  Additional Method

### B.1  Training Recipe

**TA-Tok Training**    TA-Tok is trained on both 100M raw web images and 100M aesthetic-filtered images from LAION-5B [53] to balance its ability on encoding general images for understanding and high-quality images for generation. Note we only use the images (without text) in LAION-5B. The aesthetic images are with a resolution above 512px and resized to 384px. To enable Scale-Adaptive Pooling and Decoding, we random select a scale from $\{1, 2, 3\}$, resulting in $\{729, 169, 81\}$ tokens. Since learning longer sequence is usually harder, we set the sampling ratio of scale $\{1, 2, 3\}$ to $(2 : 1 : 1)$. The training of De-Tokenizers and MLLM also follows the same sampling ratio.

**De-Tokenizer Training**    The training of De-Tokenizers is to align low-level image representations (*e.g.*, VAE [26] and VQVAE [65]) with TA-Tok's semantic representation. For AR-DTok, we train a series of models with different resolutions, ranging from 256px to 1024px. **(a) 256px.** Since AR-DTok is only developed from high-quality image generation, we train 256px AR-DTok on 50M aesthetic-filtered images from LAION. **(b) 512px.** We finetune 256px AR-DTok to 512px using 23M synthetic images (described in Sec. 3.4 Data Curation). **1024px.** To demonstrate the resolution decoupling between our TA-Tok and De-Tokenizer, we further finetune AR-DTok from 512px to 1024px using 3M synthetic images of 1024px resolution from [15]. For Dif-DTok, we directly leverage a pretrained text-to-image model, SANA-0.6B [74] as the starting model. We train Dif-DTok with 23M synthetic images at 512px. Notably, we observe that Dif-DTok achieves good convergence with just 5M training samples. However, we continue training on the full 23M dataset to ensure broader image converge and diversity.

**MLLM Prompt Format**    Since TA-Tok and MLLM are connected with discrete tokens, we expand the vocabulary of MLLM and convert TA-Tok tokens into text that MLLM can understand. The newly added vocabularies are:

```
<im_start>, <im_end>, <S0>, <S1>, <S2>, <I0>, <I1>, ..., <I65535>,
```

where `<im_start>` and `<im_end>` are the starting and end tags, `<S[0-2]>` are the scale tokens and `<I[0-65535]>` are image tokens. Following Qwen2.5-Instruct [78], the prompt of one-turn conversation is formatted as:

```
<|im_start|>user\n{Q}<|im_end|><|im_start|>assistant\n{A}<|im_end|>,
```

where `<|im_start|>` and `<|im_end|>` are the starting and end tags of an instruction (`Q`) or response (`A`). To distinguish image understanding and generation, we use different prompt format for them. For image understanding,

```
Q={text}<I1><I3><I1314>...<I520>{text}, A={text response},
```

where we do not add image start and end tags. Note the length of image tokens can be *{729, 169, 81}*, depending on the selected scale. For image generation, the prompt is:

```
Q={text prompt}, A=<im_start><I1><I3><I1314>...<I520><im_end>,
```

where we add image tags to the prompt. During inference, the generation prompt is:

```
<|im_start|>user\n{text prompt}<|im_end|><|im_start|>assistant\n<im_start>.
```

The image start tag `<im_start>` at the end of prompt encourages the model to generate image tokens instead of text tokens.

## B.2  Discussion with Related Methods

Our approach utilizes discrete visual tokens, which differs significantly from traditional continuous visual representations, such as MetaMorph [62] and MetaQuery [47]. MetaMorph constructs a visual representation by regressing SigLIP latents through an additional vision head, followed by training a diffusion decoder with an image autoencoding objective. However, this regression and reconstruction pipeline limits the ability to generate diverse images from a prompt, lacking the natural generative process seen in autoregressive language models (LLMs). In contrast, our discrete visual tokens allow direct integration with autoregressive sampling strategies, enabling diverse and controllable generation within a unified token space.

MetaQuery, on the other hand, uses a frozen multimodal LLM for condition extraction, but the image understanding and generation modules are decoupled. The understanding is handled by the frozen LLM, and the generation is handled by a diffusion decoder. This approach is closer to traditional diffusion-based methods (e.g., DiT) rather than a unified model with shared representations. In contrast, our method explicitly unifies both image understanding and generation within a single discrete semantic space (i.e., TA-Tok tokens).

The role of our De-Tokenizer also differs from that of traditional diffusion decoders [20]. Rather than generating new images conditioned on tokens, the Dif-DTok reconstructs the input image from discrete visual tokens produced by TA-Tok. These tokens already preserve essential 2D structure and visual details (e.g., a $27{\times}27$ grid of 729 tokens), and Dif-DTok's purpose is to map these structured tokens into pixel space. Therefore, we view the De-Tokenizer as a modular component within our pipeline, rather than a full generative decoder.

Furthermore, the generated visual tokens are not merely conditioning signals but 2D visual tokens that encode all necessary information for image reconstruction. As demonstrated in Appendix Tab. 11, images encoded by TA-Tok can be faithfully reconstructed by the De-Tokenizer. This is a key distinction from MetaQuery, where image generation heavily relies on a diffusion model to handle both multimodal understanding and image generation. In our approach, a simple De-Tokenizer is sufficient to convert TA-Tok's tokens into VQ-VAE or VAE tokens, eliminating the need for a complex diffusion model.

## C  Datasets

### C.1  Dataset Curation Details

The common text-to-image data collection process include a complex pipeline, *e.g.*, aesthetic filtering, watermark/logo detection and image-text alignment filtering. Instead, training with synthetic datasets is a more efficient solution [10]. We only need to focus on the quality of text prompts. The state-of-the-art generation models can provide high-quality images of a given prompt. Therefore, we develop a simple and efficient data curation pipeline:

**Image Caption**    To encourage the diversity of prompts, we use an open-source MLLM, Qwen2.5-VL-7B[3] to generate long and detailed captions for multiple datasets, including image classification datasets ImageNet-1K [13] nad ImageNet-21K [51], and a photo dataset Megalith-10M [15]. The caption prompt to Qwen2.5-VL is:

> Here are some text-to-image prompts:
>
> Example 1: {long and detailed prompt 1}
> Example 2: {prompt 2}
> Example 3: {prompt 3}
>
> Now generate a prompt for this image. Do not copy the above content. Just follow their prompt style. Do not output anything else.
> The image is: {image}

This step is similar to re-caption, but differently, we only need the text prompts generated from these images, no matter the quality of these images.

**Synthetic Image Generation** In the above step, we have obtained millions of text prompts of real world images. However, users may not input such long prompts of real world objects to the model, they often provides short, simple, and creative prompts made up of a few words, *e.g.*, *"boy, dog, the grass, happy play, 4K"*. Therefore, we also collect user prompts datasets: JourneyDB [55] and Midjourney-Prompts [66]. Although JourneyDB is a text-to-image dataset, we argue that its images are generated by very early generation models like MidJourney [*] before May, 2023. Here we adopt a state-of-the-art and fast model, FLUX.1-schnell [27] as the image generator. With only 4 sampling steps and 512px resolution, we can quickly generate a lot of images using prompts from Qwen2.5-VL and user prompts. Finally, we collect a 23M high-quality synthetic dataset.

## C.2 Unified MLLM Pretraining Data

Except the traditional pretraining tasks, *i.e,* , image-to-text (I→T), text-to-image (T→I), text-only (I→I), we also propose two new tasks: image-to-image (I→I) and text-image-to-image (TI→I).

**Image-to-Image** Given a text prompt, we ask FLUX to generate two images with different seed. This task requires the MLLM to understand the input image first and generate a similar one with the same semantic. The prompt is:

```
Q=Generate an image similar to {image 1}, A={image 2}.
```

**Image-Text-to-Image** Given a text prompt, we first split it into two parts using an open-source LLM Qwen2.5-7B [78]:

> For a given prompt, you need to replace part of its content with a placeholder <image>.
>
> For example:
> INPUT:
> In a cinematic and realistic portrayal of 1920s girls at college, we see them studying in a university with a dark academic atmosphere, where a haunting and creepy ghost story unfolds.
>
> OUTPUT EXAMPLE 1:
> {"prompt": "In a cinematic and realistic portrayal of <image>, we see them studying in a university with a dark academic atmosphere, where a haunting and creepy ghost story unfolds.", "<image>": "1920s girls at college."}
>
> OUTPUT EXAMPLE 2:
> {omitted}
>
> Now I will give you another prompt, you need to output one sample for each prompt. Just output the result in json format, and do not output anything else.

---

[*] https://www.midjourney.com

Table 7: **Dataset Summary**.

| Model | Stage | Data | Type | Size |
|---|---|---|---|---|
| TA-Tok | - | LAION (100M) [53] , LAION Aes (100M) [53] | Image | 200M |
| AR-DTok | 256px | LAION Aes (50M) [53] | Image | 50M |
| | 512px | Gen23M [13, 15, 51, 55, 66] | Image | 23M |
| | 1024px | Gen3M [15] | Image | 3M |
| Dif-DTok | 512px | Gen23M [13, 15, 51, 55, 66] | Image | 23M |
| MLLM | Pretrain | LLaVA-Recap-CC12M [39], DataComp (20M) [31] | I2T | 32M |
| | | Gen23M [13, 15, 51, 55, 66] | T2I | 23M |
| | | Gen15M [13, 15, 51, 55] | I2I, TI2I | 15M |
| | | Magpie (4M) [77], WebInstruct (12M) [80] OpenHermes [59], GenQA [7], Infinity-Instruct [45] | Text | 28.5M |
| | SFT | LLaVA-v1.5 (665K) [38], LLaVA-Recap-558K [39], LLaVA-Recap-118K [39], Self-Reflect-340K | I2T | 1.6M |
| | | Gen-SFT-1.6M | T2I | 1.6M |
| | | Magpie (1M) [77] | Text | 1M |

Table 8: **Training Parameters**.

| config | TA-Tok | AR-DTok | | | Dif-DTok | MLLM | |
|---|---|---|---|---|---|---|---|
| | - | 256px | 512px | 1024px | 512px | Prertain | SFT |
| learning rate | 2e-4 | 4e-4 | 1e-4 | 1e-4 | 1e-4 | 5e-5 | |
| lr schedule | cosine | | cosine | | constant | consine | |
| optimizer | AdamW | | AdamW | | CAME | AdamW | |
| optimizer params | $\beta_1=0.9,\beta_2=0.99$ | | $\beta_1=0.9,\beta_2=0.95$ | | $\beta_1=0.9,\beta_2=0.999$ $\beta_3=0.9999$ | $\beta_1=0.9,\beta_2=0.999$ | |
| weight decay | 1e-4 | | 0.05 | | 0.0 | 0.0 | |
| input resolution | 384 | 256 | 512 | 1024 | 512 | 384 | |
| warmup epochs | 0.04 | | 0.04 | | 0.01 | 0.03 | |
| epochs | 1 | | 1 | | 1 | 1 | |
| total samples | 200M | 50M | 23M | 3M | 23M | 100M | 4M |
| total batch size | 512 | 768 | 96 | 48 | 48 | 1024 | 256 |
| codebook loss | 1.0 | - | - | - | - | - | - |
| reconstruction loss | 1.0 | - | - | - | - | - | - |
| gradient clip | 1.0 | | 1.0 | | 0.1 | 1.0 | |
| token drop prob | - | | 0.1 | | 0.1 | - | - |
| data ratio | - | - | - | - | - | 2(und):2(gen):1(text) | |

The task is to generate an image based on a multimodal prompt. Therefore, to generate the target image, the model must understand both the text part and the image part of the multimodal prompt. At the training time, we organize the data as:

```
Q={text part 1}{image 1}{text part 2}, A={image 2}.
```

Using the same data source as Sec. C.1, we finally curate a 15M dataset for I→I and TI→I tasks.

**Relations to Other Tasks**    Our proposed tasks have a similar format of tasks like image editing and subject-driven image generation. However, their goals are different. For image editing, the goal is to modify part of an input image, following a text instruction. This task focuses on low-level image detail. For subject-driven image generation, its goal is to compose the objects of the input image into a scene, emphasizing visual consistency and preserving pixel-level details. In contrast, our task is to mitigating the modality gap between visual understanding and generation. Rather than persevering exact pixel information, we focus on capturing and conveying the semantic content of the input image.

## C.3 Dataset Summary

We summary all the training data in Tab. 7.

# D   Additional Implement Details

We list the training hyper-parameters in Tab. 8

# E   Additional Ablation Experiments

**Ablation of Text-Aligned Codebook**   As shown in Tab. 9, we compared different settings of TA Codebook. Tab. 9 (a) is a conventional setting, with random initialized, low dimension codebook; Tab. 9 (b) increases the codebook dimension to 1536, but fails to converge during training; Tab. 9 (c) is our LLM embedding aligned codebook with large vocabulary and high dimension. The results indicate that our TA Codebook (c) outperforms the conventional codebook (a) by a large margin in understanding tasks and is comparable on the generation task.

Table 9: **Ablation of Different Codebook Settings.**

| setting | init | size | dim | GQA | MME | POPE | DPG |
|---------|------|------|-----|-----|-----|------|-----|
| (a) | random | 32768 | 16 | 59.4 | 1261 | 86.7 | 70.9 |
| (b) | random | 32768 | 1536 | | not converge | | |
| (c) | LLM | 65536 | 1536 | **61.0** | **1428** | **87.4** | 70.6 |

**Self Reflect**   Since a unified MLLM can handle both image understanding and generation tasks, we leverage its understanding ability to assess the generation quality, *i.e.*, image-prompt alignment, which is relative simple task for modern MLLMs. Therefore, we build Self-Reflect-340K, a dataset to judge image-prompt alignment. We collect a set of prompts in the style of GenEval and DPG Bench, and query Qwen2.5-VL-7B to generate judgments. In Tab. 10, the model trained with Self Reflect show consistent improvements on the visual generation task. Notably, weaker models benefit more: the relative improvements are 0.04 for 10K-step model, but 0.016 for 60K-step model.

Table 10: **Ablation of Self Reflect.** The evaluation metric is GenEval overall score.

| Train step | 10K | 20K | 30K | 40K | 50K | 60K |
|------------|-----|-----|-----|-----|-----|-----|
| baseline | 0.717 | 0.747 | 0.753 | 0.757 | 0.759 | 0.764 |
| w/ Self Reflect | 0.757 | 0.779 | 0.788 | 0.784 | 0.785 | 0.780 |
| Δ | +4.0% | +3.2% | +3.5% | +2.7% | +2.6% | +1.6% |

Table 11: **Performance on ImageNet 256×256 Benchmark.** FID: Frechet inception distance. IS: inception score. cfg: classifier-free guidance.

| Model | #Param | cfg | FID↑ | IS↑ | Precision↑ | Recall↑ |
|-------|--------|-----|------|-----|------------|---------|
| Llamagen-Tok | | | 2.19 (rFID) | | | |
| Llamagen-XL | 775M | 1.75 | 2.62 | 244.08 | 0.80 | 0.57 |
| Llamagen-XXL | 1.4B | 1.75 | 2.34 | 253.90 | 0.80 | 0.59 |
| TA-Tok+AR-DTok | 1.2B | 10.0 | 2.60 | 208.46 | 0.76 | 0.64 |

**Image AutoEncoding Performance of TA-Tok**   Our TA-Tok and de-tokenizers can be viewed as an image autoencoding pipeline, where an input an image is encoded and then reconstructed. Since our de-tokenizers are based on generative models (AR or diffusion), one concern is whether they can faithfully recover the original image. In Tab. 11, we compare our method against LLamagen's tokenizer (Llamagen-Tok), Llamagen-XL/XXL on the ImageNet 256×256 benchmark. Note that we take TA-Tok's output as condition and Llamagen-XL/XXL are conditioned on class labels, so the results are not directly aligned. Anyway, this serves as a comparison to assess the autoencoding

capability of our method. Our method achieves comparable FID compared with Llamagen-XL, demonstrating effective reconstruction quality. We also observe that our model requires a higher classifier-free guidance (10.0) compared with Llamagen's 1.5, indicating a stronger reliance of condition.

**Compositional Generation Emerges from Unified Pretraining Tasks** In Sec. 3.4, we propose unified pretraining with two new tasks: image-to-image (I→I) and text-image-to-image (TI→I), which encourages multimodal prompt conditioned generation. As shown in Fig. 7, Tar demonstrates emergent subject-driven generation and reference-based style transfer.

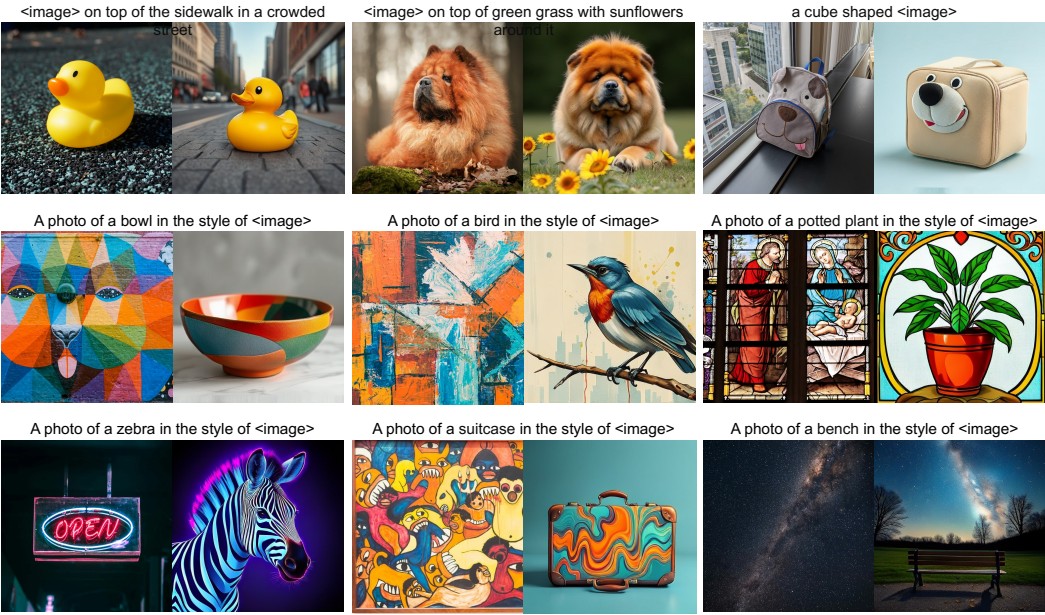

Figure 7: **Emergent Compositional Generation Ability.** The first row denotes subject-driven generation. The middle and bottom rows are reference-based style transfer. For each example, the left image is `<image>`, the right image is the generated image.

## F    Additional Main Results

We list the full visual generation results in Tab. 12 and Tab. 13.

## G    More Visualization

We give more visualization of our model on the text-to-image generation task, shown in Fig. 8. Notably, our model can follow Chinese prompts even it is not trained with Chinese prompt dataset, by leveraging the multilingual ability of the base LLM, Qwen2.5 [78].

## H    Limitation and Future Work

Our method has several limitations. First, the vector quantization in TA-Tok introduces quantization errors. Despite mitigating this with a larger codebook and extended training, some information loss remains—particularly in tasks requiring fine-grained understanding, such as OCR. This could be improved by adopting longer visual sequences and incorporating techniques like Token-Shuffle [44]. Second, while our method excels at generating diverse images, it underperforms in accurately reconstructing the input image compared to traditional tokenizers [65]. This limitation stems from using generative models as de-tokenizers. A potential solution is to train the de-tokenizer as an image super-resolution model to better preserve local consistency between the input and reconstructed images.

# I Societal Impact

This work advances unified MLLMs by introducing a shared, discrete semantic representation that enhances both visual understanding and generation, enabling efficient language-vision integration for applications like assistive tools, creative content, and education.

However, high-quality image generation also poses risks, including potential misuse for misinformation or manipulation. While our model is not identity-specific, downstream use should include safeguards such as watermarking and prompt filtering. We advocate for ethical use, emphasizing fairness, robustness, and transparency.

Table 12: **Visual Generation Results on GenEval [22].**

| Method | Single Obj. | Two Obj. | Counting | Colors | Position | Color Attri. | Overall↑ |
|---|---|---|---|---|---|---|---|
| *Generation Only Model* | | | | | | | |
| SDv1.5 [52] | 0.97 | 0.38 | 0.35 | 0.76 | 0.04 | 0.06 | 0.43 |
| PixArt-$\alpha$ [9] | 0.98 | 0.50 | 0.44 | 0.80 | 0.08 | 0.07 | 0.48 |
| SDv2.1 [52] | 0.98 | 0.51 | 0.44 | 0.85 | 0.07 | 0.17 | 0.50 |
| Emu3-Gen [69] | 0.98 | 0.71 | 0.34 | 0.81 | 0.17 | 0.21 | 0.54 |
| SDXL [48] | 0.98 | 0.74 | 0.39 | 0.85 | 0.15 | 0.23 | 0.55 |
| DALLE3 [36] | 0.96 | 0.87 | 0.47 | 0.83 | 0.43 | 0.45 | 0.67 |
| SD3-Medium [16] | 0.99 | 0.94 | 0.72 | 0.89 | 0.33 | 0.60 | 0.74 |
| SANA-1.5 [75] | 0.99 | 0.93 | 0.86 | 0.84 | 0.59 | 0.65 | 0.81 |
| *Unified Model* | | | | | | | |
| Chameleon-7B [58] | - | - | - | - | - | - | 0.39 |
| LWM-7B [37] | 0.93 | 0.41 | 0.46 | 0.79 | 0.09 | 0.15 | 0.47 |
| SEED-X-13B [21] | 0.97 | 0.58 | 0.26 | 0.80 | 0.19 | 0.14 | 0.49 |
| Show-o-1.3B [76] | 0.95 | 0.52 | 0.49 | 0.82 | 0.11 | 0.28 | 0.53 |
| Transfusion-7B [84] | - | - | - | - | - | - | 0.63 |
| D-DiT-2B [35] | 0.97 | 0.80 | 0.54 | 0.76 | 0.32 | 0.50 | 0.65 |
| ILLUME-7B [67] | 0.99 | 0.86 | 0.45 | 0.71 | 0.39 | 0.28 | 0.61 |
| Janus-1.3B [70] | 0.97 | 0.68 | 0.30 | 0.84 | 0.46 | 0.42 | 0.61 |
| Janus-Pro-1B [10] | 0.98 | 0.82 | 0.51 | 0.89 | 0.65 | 0.56 | 0.73 |
| Harmon-1.5B [72] | 0.99 | 0.86 | 0.66 | 0.85 | 0.74 | 0.48 | 0.76 |
| Janus-Pro-7B [10] | 0.99 | 0.89 | 0.59 | 0.90 | 0.79 | 0.66 | 0.80 |
| **Tar-1.5B$^*$ (Ours)** | 0.99 | 0.91 | 0.76 | 0.81 | 0.57 | 0.51 | 0.76 |
| *w/ Self Reflect* | 0.99 | 0.92 | 0.77 | 0.81 | 0.62 | 0.55 | 0.78 |
| **Tar-7B$^*$ (Ours)** | 0.98 | 0.92 | 0.83 | 0.85 | 0.80 | 0.65 | 0.84 |
| *w/ Self Reflect* | 0.98 | 0.93 | 0.86 | 0.85 | 0.80 | 0.70 | **0.85** |

Table 13: **Visual Generation Results on DPG Bench [24].**

| Method | Global | Entity | Attribute | Relation | Other | Overall↑ |
|---|---|---|---|---|---|---|
| *Generation Only Model* | | | | | | |
| SDv1.5 [52] | 74.63 | 74.23 | 75.39 | 73.49 | 67.81 | 63.18 |
| PixArt-$\alpha$ [9] | 74.97 | 79.32 | 78.60 | 82.57 | 76.96 | 71.11 |
| Emu3-Gen [69] | 85.21 | 86.68 | 86.84 | 90.22 | 83.15 | 80.60 |
| SDXL [48] | 83.27 | 82.43 | 80.91 | 86.76 | 80.41 | 74.65 |
| Playground v2.5 [29] | 83.06 | 82.59 | 81.20 | 84.08 | 83.50 | 75.47 |
| Hunyuan DiT [34] | 84.59 | 80.59 | 88.01 | 74.36 | 86.41 | 78.87 |
| PixArt-$\Sigma$ [8] | 86.89 | 82.89 | 88.94 | 86.59 | 87.68 | 80.54 |
| DALLE3 [36] | 90.97 | 89.61 | 88.39 | 90.58 | 89.83 | 83.50 |
| SD3-Medium [16] | 87.90 | 91.01 | 88.83 | 80.70 | 88.68 | 84.08 |
| SANA-1.5 [75] | - | - | - | - | - | **84.70** |
| *Unified Model* | | | | | | |
| Janus-1.3B [70] | 82.33 | 87.38 | 87.70 | 85.46 | 86.41 | 79.68 |
| Janus-Pro-1B [10] | 87.58 | 88.63 | 88.17 | 88.98 | 88.30 | 82.63 |
| Janus-Pro-7B [10] | 86.90 | 88.90 | 89.40 | 89.32 | 89.48 | 84.19 |
| **Tar-1.5B$^*$ (Ours)** | 83.59 | 89.35 | 86.91 | 93.50 | 80.80 | 82.96 |
| *w/ Self Reflect* | 84.17 | 88.48 | 87.83 | 93.38 | 84.07 | 84.10 |
| **Tar-7B$^*$ (Ours)** | 83.98 | 88.62 | 88.05 | 93.98 | 84.86 | 84.19 |
| *w/ Self Reflect* | 84.09 | 88.60 | 88.78 | 93.59 | 85.15 | 84.65 |

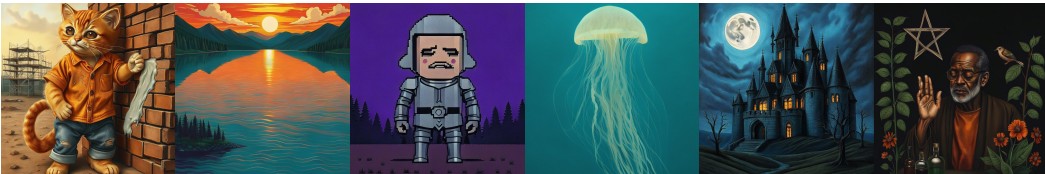

1. An illustration of an ultra-detailed anthropomorphic orange kitten wearing a patched, oversized construction worker t-shirt and rolled-up muddy jeans. The kitten stands on hind paws, holding a masonry brush, applying glossy wet cement to a red brick wall. The background shows an unfinished construction site with scaffolding, scattered tools (shovel, wheelbarrow), wooden planks, and piles of bricks. The kitten has large cartoonish eyes with a determined expression, fluffy fur with dirt smudges, and a tiny cement splatter on its cheek. Soft golden-hour lighting, warm vibrant colors, hyper-realistic textures, 8K resolution, trending on ArtStation.
2. bright colorful illustration of lake with view mountain and hype detailed sunset view
3. 8-bit Pixel art, square bog pixels, flat colors, sad soldier with silver armor, purple sky with dark forest background
4. highly detailed full body jellyfish
5. nighttime view of a gothic horror castle with vivid colors and a wood press style, dark fantasy
6. African spiritual guide with a gentle air, green plants, very colorful flowers and perfume bottles on a black background. His fetish object is a metal rod with seven points pointing towards the sky and a bird on the central rod.

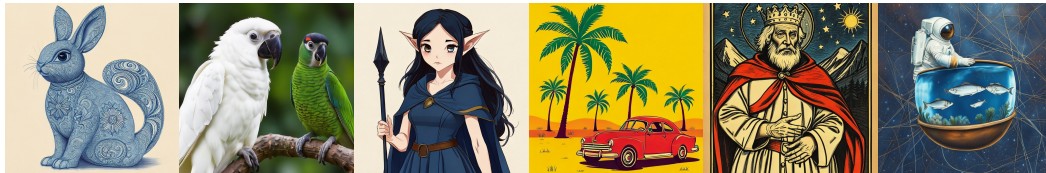

1. a chinese rabbit made of white ceramic with blue ink, stunning intricate designs, ((geometric and floral patterns, fine art))
2. Fischer's lovebird, Galah Cockatoo, Lutino Ringneck Parakeet and other psittacidae species.
3. Create a fusion of Frieren from Frieren: Beyond Journey's End and Sailor Moon from Sailor Moon, with the character primarily resembling Frieren. She should have Frieren's calm and timeless expression, pointed elf ears, and signature pale blue hair styled in Sailor Moon's iconic twin buns with flowing strands. Her attire should merge Frieren's mage-like robes with subtle Sailor Moon-inspired celestial details, such as crescent moons and stars integrated into the design. She should wield a staff reminiscent of Frieren's wand, enhanced with a crescent moon motif at the top. Surround her with a soft, ethereal glow, symbolizing the fusion of Frieren's wisdom and magical mastery with Sailor Moon's celestial grace and heroism.
4. few palm tree and an old car, simple background, pop art style
5. Wiccan tarot. Emperor. sketch, color ink, thin outline, comic art
6. Astronaut with fish tank is moving inside the galactic space suspended in the air with very fine and intertwined lines and acid watercolor and oil colors with strong contrast with different details and colors.

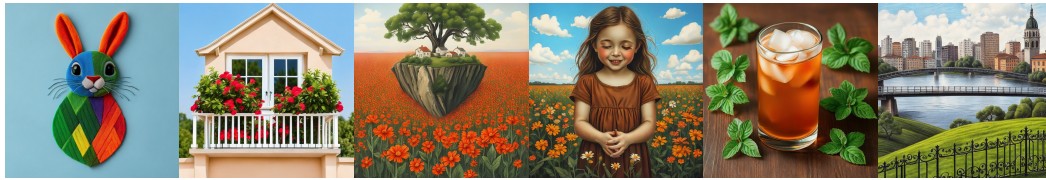

1. a female rabbit head, minimalistic colorful organic forms, energy, assembled, layered, depth, alive vibrant, 3D, abstract, on a light blue background
2. Blooming balcony goals. This house boasts a beautiful balcony overflowing with colorful flowers.
3. A fully detailed city on a large tree in the middle of a plain full of red and pink flowers and pieces of rock
4. A girl who has drawn a beautiful heart with the clouds in the sky in a field full of flowers with her hands
5. Fresh sweet iced tea is displayed on the table with mint flowers
6. Beautiful pen and ink sketch of San Francisco, photorealist, colored, detailing, daytime

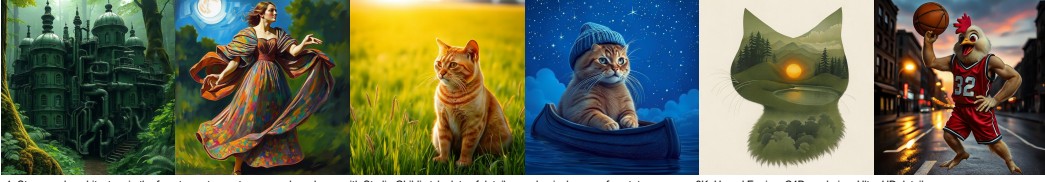

1. Steampunk architecture in the forest, reactor, rusty green color scheme with Studio Ghibli style, lots of details, mechanical, green, forest, trees, moss, 8K, Unreal Engine, C4D rendering, Ultra HD details.
2. Painterly Style, mixture of oil paints and acrylic paints and watercolors, clear evidence of the different paint mediums, Style of Vincent Van Gogh, Style of Renoir, Style of Claude Monet, Style of Pierre Bonnard, Style of Camille Pissaro, Style of Paul Cezanne, a stunning dancer wearing a long and flowing gown with vibrant colors, she is wearing a shawl with delicate embroidery, the background is a garden in the moonlight
3. A serene photograph of a ginger and white cat sitting in a sunlit grassy field. The cat is positioned slightly to the right, gazing upwards with a calm expression. Its fur is a soft orange with distinct white patches on its chest and face. The foreground features out-of-focus blades of grass, creating a dreamy bokeh effect. The background is a blurred mix of soft greens and browns, suggesting a natural outdoor setting. The lighting is warm and golden, highlighting the cat's fur and casting gentle shadows. The image has a shallow depth of field, emphasizing the cat while the background remains softly blurred. Photorealistic, tranquil, natural lighting, warm color palette, high contrast, intimate, peaceful atmosphere.
4. 微缩景观,毛茸茸羊毛毡,超级特写,浅景深,梵高风格的星空下,一只小猫咪坐在一艘发光的小船上,船的周围漂浮着毛茸茸的羊毛毡星星,猫咪头顶戴 着一顶星光点缀的小帽子,背景是旋转的漩涡星空,散发着梦幻的蓝金光芒,生物发光.细节丰富,3D立体.
5. surreal art of landscape morphing into a cat's head silhouette, intricate nature elements forming feline features, mountains create pointed ears, rolling hills shape the face curves, forests and meadows texture the fur pattern, rivers flow as whiskers, lakes form the eyes with sunset reflection, clouds define soft fur edges, minimal white background highlighting the cat head shape, dreamlike composition, delicate details, organic flowing lines, nature meets feline anatomy, soft muted color palette, ethereal atmosphere, conceptual illustration style, fine art quality.
6. hyperrealistic 3D render of a stylized cartoon chicken in basketball uniform, dynamic slam dunk pose, jumping high with basketball, urban street court background, dramatic action shot, motion blur effects, neon lights at dusk, streetball atmosphere, detailed feather texture, glowing court lines, floating movement, cinematic sports photography style, vibrant complementary colors.

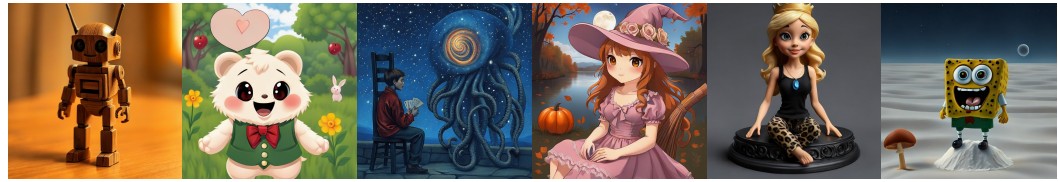

1. The image showcases a meticulously crafted wooden robotic figure. The robot is intricately designed with multiple components, including a head with two long protruding antennae, a torso with various compartments, arms, and legs. The figure is displayed on a wooden surface, and the background is blurred, emphasizing the robot as the main subject. The ambiance suggests a calm and serene environment, possibly a room with ambient lighting.
2. This image showcases a cartoon character, possibly from a video game, set in a lush, green park-like environment. The character has a cute appearance with a yellow and white fur coat, large eyes, and a cheerful expression. The character is wearing a green vest with a white shirt underneath, and a red bowtie. Above the character, there's a speech bubble with a heart symbol inside it. In the background, there are trees, cherries, and a small white rabbit-like creature. The overall ambiance of the image is joyful and vibrant.
3. This image showcases a surreal scene where a boy in a red jacket sits on a wooden chair, holding a fan of cards. Beside him is a colossal cosmic creature with a blue body featuring a swirling galaxy-like pattern and numerous tentacles. The backdrop is a star-filled night sky, creating an otherworldly and thought-provoking atmosphere.
4. The image showcases an animated character, presumably a female, with long orange hair and golden eyes. She is wearing a pink dress adorned with lace and a large pink witch hat decorated with flowers and a pumpkin. The character is surrounded by a magical, autumnal setting. There are pumpkins, a carved pumpkin face, and a broomstick. The background features a serene lake with a full moon shining brightly, surrounded by trees and decorated with colorful lights.
5. This image showcases a 3D-rendered character, possibly a female, with blonde hair and blue eyes. She is adorned with a crown on her head and wears a black top with a blue gem pendant. The character is seated on a circular base with intricate designs, and she has leopard print pants. The overall color palette is vibrant, with a mix of blues, blacks, and the warm tones of the leopard print.
6. 3D-rendered SpongeBob SquarePants on a desert-like sand mound at night. He has his typical yellow square body with pores, large green-eyed, open-mouthed look, wearing a white shirt, red tie, green shorts, striped socks and black shoes. A brown mushroom is nearby, with a star-filled sky and a pale moon in the background.

Figure 8: **Visual Generation Results.** We use Tar-7B and 1024px AR-DTok to generate these images. Most prompts are from the web, and a few prompts are from [49, 75]. Zoom in for better view.

