# OpenReview forum: "Vision as a Dialect: Unifying Visual Understanding and Generation via Text-Aligned Representations"
_NeurIPS.cc/2025/Conference — NeurIPS 2025 poster_

### Official Review · Reviewer_7n9v · 2025-06-29

**Clarity:** 1
**Significance:** 3
**Originality:** 2
**Rating:** 4
**Confidence:** 3

**Summary:**

The paper introduces Tar, a framework to integrate visual understanding and generation within a unified architecture, operating on discrete tokens. The main ingredient is Text-Aligned Tokenizer (TA-Tok), which projects images into a discrete codebook of text-aligned tokens through a trainable SigLIP2 encoder + quantization. By pairing this tokenization strategy with either autoregressive or diffusion-based de-tokenizers, Tar demonstrates strong performance across a variety of benchmarks for both image understanding and generation tasks.

**Questions:**

- The paper claims that the codebook is semantically aligned. However, since the projection matrix W is trainable, is it possible that this alignment is compromised? How is semantic alignment preserved?
- What is the criterion used to select the top-k most representative embeddings when constructing the visual codebook?
- In Table 2, individual best-performing scores are not highlighted. Would the authors consider emphasizing these for easier comparison?

**Ethical Concerns:**

["NO or VERY MINOR ethics concerns only"]

**Final Justification:**

I believe this paper is technically sound and presents a promising approach to unified text-image understanding and generation, demonstrating strong results across multiple benchmarks. My main concerns lie in the clarity of the manuscript; however, the authors have addressed most of these issues in their response and have committed to improving the presentation in the final version. For these reasons, my overall assessment is borderline, but I am leaning toward acceptance.

**Limitations:**

Limitations and societal impact are discussed in the supplementary material.

**Paper Formatting Concerns:**

No concern.

**Quality:**

3

**Strengths And Weaknesses:**

**Strengths:**

- TA-Tok provides a novel approach to image tokenization for multimodal LLMs. The idea of pairing the semantic richness of continuous tokens produced by a fine-tuned encoder (SigLIP2) and the advantages of discrete tokens for image generation is interesting and promising for future architectures.

- Tar exhibits competitive or state-of-the-art performance across various benchmarks, both in the image generation and image understanding domains.

**Weaknesses:**

I believe this work is solid on the technical side, but it would greatly benefit from improvements in clarity and presentation. Several aspects are currently difficult to follow or underexplained:

- **LLM training strategy.** It is unclear whether the backbone LLM is trained from scratch or fine-tuned. Section 3.4 suggests that it is pre-trained and then fine-tuned, while line 225 mentions initialization from Qwen2.5-Instruct. This is further confused by the subsequent statement that the LLM is 'fully finetuned at both the pretraining and finetuning stage'. Clarification on the training pipeline would be helpful.

- **Scale Adaptive Pooling and Decoding.** These components are listed as key contributions of the paper, yet their mechanisms are not described in sufficient detail. A more thorough explanation of how these modules work and of their specific role within the overall architecture would strengthen the paper.

- **Comparison with other visual representations (Section 4.3).** The setup of this experiment is somewhat opaque. It would be useful to clarify: i) whether all methods are trained on the same data ii) how the authors isolate the impact of visual tokenization from the influence of different decoding strategies, which also vary across configurations.

- **Pre-training strategies.** The I->I and TI->I strategies introduced in Section 3.3 are rather unclear, and a more detailed explanation would be valuable. Specifically, for TI->I the example does not really seem to help: what is the role of the second chunk of sentence ('the grass')?

- **Figure 4.** The figure illustrating the two decoding strategies is somewhat confusing. It is not immediately clear how the de-tokenizers relate to the autoregressive LLM at inference time. The direct connection from TA-Tok to the decoding modules, and the presence of an autoregressive model in only one of the two pathways, makes the architectures difficult to interpret.

- **Self Reflect.**  The authors briefly discuss the introduction of a 'Self Reflect strategy', which improves the generation results. I believe some hints on its internals should be provided in the main text (or, at least, a pointer to the relevant appendix section).

**Typos:**

The manuscript contains some minor typos, including:

- line 144 and 188 ('to decoding' -> 'to decode')
- line 157 ('vis' -> 'via')
- line 168 ('particular' -> 'particularly')
- lines 205-206 ('text-image to text' should probably be 'text-image to image')
- line 226 ('random' -> 'randomly')
- line 233 ('a fully discrete' -> 'fully discrete')
- lines 274-275 (meaning is clear but sentence structure needs revision)
- line 277 ('performs' -> 'perform')
- lines 282-283 ('makes our module can produce' needs revision)

---

> ### Author Rebuttal · Authors · 2025-07-31
>
> Dear Reviewer 7n9v,
>
> Thank you for your detailed comments!
>
> **Q1: LLM training strategy. It is unclear whether the backbone LLM is trained from scratch or fine-tuned.**
>
> Ans: In Sec.3.4, we follow previous works on multimodal LLMs [2,31,35] and Unified MLLMs [6,55], which typically divide the training pipeline into two stages: Pretraining and Supervised Finetuning.
>
> In this context, Pretraining refers to training the multimodal LLM, initialized from a **pretrained LLM backbone**. Specifically, we initialize our model using Qwen2.5-Instruct, a pretrained LLM, and then conduct MLLM Pretraining followed by Supervised Finetuning. We will clarify the difference between LLM and MLLM pretraining in the revision.
>
>
> **Q2: More explanation of Scale Adaptive Pooling and Decoding**
>
> Ans: **(a) Mechanism:** As described in **Figure 3 (caption), Lines 122–129, 130–140 (especially Lines 131–132), and Lines 226–227**, Scale-Adaptive Pooling (SAP) is implemented using 2D adaptive pooling with a predefined scale set {1, 2, 3}. During training, a scale is randomly sampled; at inference, a fixed scale is used. For a 384×384 input image, these scales produce 729 (27×27), 169 (13×13), and 81 (9×9) tokens, respectively.
> To support variable token lengths, we introduce Scale-Adaptive Decoding (SAD): a lightweight ViT decoder (identical to SigLIP2’s ViT, see **Lines 127–129**) that can process variable-length sequences by dynamically resizing its 2D positional embeddings.
>
> **(b) Role:** SAP and SAD allow TA-Tok to extract multi-granularity features, as noted in Lines 122–125. This flexibility enables the MLLM to better adapt to different tasks:
> - Coarse-grained features (e.g., 81 tokens) are suitable for text-to-image generation, especially under limited training data.
> - Fine-grained features (e.g., 729 tokens) are crucial for detailed image understanding tasks.
> This design is empirically validated in **Table 5**: visual understanding tasks consistently benefit from more tokens, while generation tasks prefer fewer tokens in low-data regimes, but improve with more tokens when training data is scaled up.
>
> **Q3: Comparisons to other visual representations.**
>
> **(1) Whether all methods are trained on the same data?**
>
> Ans: Yes. Line 250-252: “For training MLLMs with these representations, we sample **a subset of our training data for controlled experiments**: 10M T2I data, 10M I2T data and 5M text-only data. All models are **trained with the same configuration** and tested on visual understanding [15, 20, 22, 26] and generation tasks [19].”
>
> **(2) How the authors isolate the impact of visual tokenization from the influence of different decoding strategies.**
>
> In Section 4.3, we compare TA-Tok with other discrete visual tokenization methods, including VQ-VAE, Janus (which also uses VQ-VAE for image generation), and a hybrid tokenizer (i.e., UniTok [36]). All these methods adopt an **encoder–decoder architecture**, where the encoder (tokenizer) converts input images into discrete tokens, and the decoder (de-tokenizer) reconstructs the image from those tokens.
> We understand that the reviewer may question whether our decoding setup is fair compared to the baselines. However, we would like to emphasize that the proposed **de-tokenizer (AR-DTok and Dif-DTok) is one of the key contributions** of our paper. They are specifically designed to decode **fully semantic discrete tokens** (produced by TA-Tok) back to pixel-level images, using autoregressive and diffusion-based models, respectively. Therefore, while decoding strategies do differ across methods, these differences are **an integral part of the overall design and contribution** of our approach, not confounding factors in the comparison.
>
> **Q4: Pre-training strategies**
>
> **(1) More detailed explanation of the proposed I2I and TI2I tasks.**
>
> Ans: As described in **Sec. 3.4 and Appendix Sec. D.2**, the proposed I2I and TI2I tasks encourage the MLLM to first understand the input image and then generate another image, which means the model needs to improve its understanding and generation task in one training sample. We will add more details for the two tasks.
>
> **(2) "A dog running on the grass" becomes: "A dog running on \<image\>" and "the grass". Explain this example.**
>
> Ans: As described in **Line 206-208 and Appendix Sec. D.2**, we use FLUX to generate **two** images based on the prompts “A dog running on the grass” and “the grass”. The image with prompt “the grass” is used as an input image. The task is to generate a final image based on a **multimodal prompt**: “A dog running on \<image\>”.
>
> **Q5: The two decoding strategies in Figure 4**
>
> **(1) How the two-de-tokenizers relate to the autoregressive LLM at inference time?**
>
> Ans: As described in **Figure 2 and Sec. 3.3 Inference (Line 186-188)**, the LLM first generates a sequence of visual tokens during inference. These tokens are then passed to either AR-DTok or Dif-DTok to reconstruct the final image. In other words, the de-tokenizers operate **after the LLM** and are responsible for converting the predicted tokens back to the pixel space.
>
> **(2) Why does the autoregressive model only appear in one of the two decoding pathways?**
>
> Ans: The “Autoregressive Model” shown in Figure 4(a) refers **specifically** to the **image generation model used within AR-DTok**, which is a separate autoregressive model trained to reconstruct images from visual tokens. It is not the same as the autoregressive LLM depicted in Figure 2. To clarify: (a) Both AR-Dtok and Dif-DTok receive the LLM-generated tokens at inference time. (b) During training, they both consume tokens from TA-Tok.
>
> **Q6: Self Reflect requires more details in the main text**
>
> Ans: Due to the space limit, we mainly introduce the Self Reflect strategy in Appendix Sec. F and its ablation experiments. We will add more detail of Self Reflect in the main text and give a link to the appendix for more details.
>
>
> **Q7: Since the projection matrix W in Text-Aligned Codebook is trainable, is it possible that this alignment is compromised?**
>
> Ans: As shown in Figure 3, we initialize the codebook using **frozen LLM embeddings** from a text-pretrained LLM. This provides a **strong semantic prior**: each token in the codebook corresponds to a position in the LLM’s text space. So even though the projection is learned, it must learn to **align with a fixed semantic space**. This constraint naturally preserves semantic consistency during training.
>
>
> **Q8: What is the criterion used to select the top-k most representative embeddings when constructing the visual codebook?**
>
> Ans: As described in Line 120-121 and the following code, we select the top-k most representative embeddings based on their average distance to others.
> ```
> # embeddings.shape=[150K, D]
> norms = torch.norm(embeddings, dim=1, keepdim=True)
> cosine_sim = torch.matmul(embeddings, embeddings.T) / (norms * norms.T)
> average_similarity = torch.mean(cosine_sim, dim=1)
> topk=65536
> representative_tokens = torch.argsort(average_similarity)[:topk]
> ```
>
> **Q9: In Table 2, individual best-performing scores are not highlighted.**
>
> Ans: We will highlight them in the revision.

---

> > ### Comment · Reviewer_7n9v · 2025-08-05
> >
> > Thank you for the detailed response to my concerns. Most of them were resolved, and I believe that incorporating more detailed explanations in the paper could greatly benefit future readers. I am considering updating my score.
> >
> > Regarding the point referenced as Q3, I understand and acknowledge that the de-tokenizer is a crucial contribution of your proposed method, and its strengths should not be seen as a confounding factor. That said, I believe the title of the section “Comparisons with Other Visual Representations” could be misinterpreted, as the differences are not limited to the representation alone, but also (rightfully) include the decoding process.

---

> > > ### Author Response · Authors · 2025-08-05
> > > **Grateful for Your Recognition and Suggestions**
> > >
> > > Dear Reviewer 7n9v,
> > >
> > > Thank you very much for recognizing the value of our reply and your kind consideration of updating the score. We’re truly grateful that most of your concerns have been resolved.
> > >
> > > We fully agree with your point regarding the section title “Comparisons with Other Visual Representations”. The current phrasing may indeed unintentionally understate the role of the decoding process. We will revise both the title and relevant text to better reflect the scope of the comparison.
> > >
> > > Once again, we sincerely appreciate your constructive feedback throughout the review process, especially your helpful comments on the implementation details, typos, and overall presentation of the paper. Your suggestions have been very helpful in improving the quality of our work.
> > >
> > > Best regards,
> > >
> > > The Authors

---

> ### Author Response · Authors · 2025-08-04
> **Gentle Reminder**
>
> Dear Reviewer 7n9v,
>
> We sincerely appreciate the time and effort you have dedicated to reviewing our paper. In response to your comments, we have provided detailed clarifications to address your specific concerns.
>
> As the discussion period will conclude in a few days, we would greatly appreciate it if you could kindly take a moment to review our rebuttal at your convenience. If there are any remaining questions or suggestions, we are more than happy to address them promptly.
>
> Thank you once again for your thoughtful feedback and valuable contribution to our work.
>
> Best regards,
>
> The Authors

---

> ### Author Response · Authors · 2025-08-07
> **Follow-up on Score Update**
>
> Dear Reviewer 7n9v,
>
> Thank you again for your thoughtful and constructive feedback.
>
> As the discussion period is coming to a close, we were wondering if you have had a chance to consider **updating your score**. Your active participation and perspective are **incredibly valuable to the final assessment** of our work, and we would be grateful for your continued input.
>
> Best regards,
>
> The Authors

---

### Official Review · Reviewer_GJa8 · 2025-07-02

**Clarity:** 3
**Significance:** 3
**Originality:** 3
**Rating:** 5
**Confidence:** 3

**Summary:**

This work introduces a unified multimodal framework that bridges visual understanding and generation through a shared discrete semantic representation. The core innovation is from TA-ToK, which converts images into discrete tokens using an aligned codebook projected from an LLM's vocabulary embedding space. The system employs scale-adaptive encoding/decoding for flexible visual detail control and complementary generative AR-DTok/Diff-DTok for high-quality image synthesis. Experimental results demonstrate that Tar achieves competitive performance on both visual understanding benchmarks and generation tasks, matching or surpassing specialized models while maintaining a unified architecture.

**Questions:**

1. In Figure 6 (right), AR-DeTok shows better capability in generating complete and correct text within images. The authors are encouraged to provide insights into this phenomenon.
2. Line 282: DPG is designed for evaluating T2I performance. It would be beneficial to demonstrate the impact of SAP design for I→I and TI→I scenarios.

Please also check the "weakness" section.
I am willing to change my score if my concerns are addressed.

**Ethical Concerns:**

["NO or VERY MINOR ethics concerns only"]

**Final Justification:**

I thank the authors for their response to the review comments and for conducting complementary experiments on the SAP design for TI2I tasks. I have also read the authors' responses to feedback from other reviewers. Most of my concerns have been adequately addressed.

Regarding the TI2I task, while the authors prioritize semantic alignment over finer-grained consistency, this focus could potentially limit the model's applicability in light of recent advancements in unified MLLMs. Nevertheless, the concept of an aligned codebook across modalities remains compelling and shows some promise.

Taking these factors into account, my final assessment falls between 'borderline accept' and 'accept'. I raise my score to 'accept' on the condition that the authors adequately incorporate the discussed points and complementary experiments into the final version of the manuscript.

**Limitations:**

Yes

**Quality:**

3

**Strengths And Weaknesses:**

Strengths:
1. The model has an elegant unified architecture through text-aligned discrete representations.
2. The comprehensive experimental design demonstrates strong empirical performance across both understanding and generation tasks.


Weaknesses:
1. In Table 6, joint training provides much higher gains on generation tasks than understanding tasks. The authors are encouraged to provide intuition for this imbalance. This observation suggests that image generation contributes little to visual understanding.
2. Text + multiple images → image is a practical scenario (e.g., generating the next story frame based on description and prior images). While the authors mention related tasks in Appendix lines 102-109, the boundary between such text+images-to-image and the proposed TI→I remains vague. It would be beneficial if the authors could provide guidance for this application, such as prompt format design.
3. Several images in Figure 7 in the appendix do not align well with the prompts. For example, row 3-4, 3-6, 5-4.

---

> ### Author Rebuttal · Authors · 2025-07-31
>
> Dear Reviewer Gja8,
>
> Thank you very much for your insightful comment!
>
>
> **Q1: Add intuition for the imbalance gain of understanding and generation tasks in Tab.6.**
>
> Ans: Thanks for your valuable suggestion. We will clarify this point in the revision. We believe the observed imbalance arises from two main factors:
>
> **(a) Task and supervision mismatch.** Generation tasks directly benefit from joint training because they require the model to **understand the concept (prompt)**-whether text or image-before generating coherent images. In this sense, improvements in visual understanding directly enhance generation quality.
>
> In contrast, understanding tasks are often **discriminative** in nature (e.g., classification or multiple choice), which **do not directly benefit** from generation supervision. For example, predicting visual token sequences may not significantly help with answering a Yes/No question.
>
>
> **(b) Asymmetry in task difficulty and data requirements**: The model’s **understanding ability is already relatively strong**, often requiring less data to achieve good performance. For example, models like LLaVA reach strong image understanding with only 0.6M training samples. In contrast, image generation is a **more complex task** that requires substantially more data (often 10M+ samples) and benefits more from improved multimodal representations.
> As a result, joint training helps generation more noticeably—since it benefits from enhanced visual understanding and alignment. Meanwhile, understanding tasks gain less from generation supervision, as their performance tends to **saturate earlier**, and the benefits from joint training **diminish** once the understanding capability reaches a certain level.
>
>
> **Q2: The relation between the proposed TI2I task to traditional TI2I task.**
>
>
> Ans: Traditional TI2I tasks, such as image editing, subject-driven generation, or next-frame prediction, primarily emphasize **pixel-level consistency**. For example, image editing requires unedited regions unchanged, while subject-driven generation ensures the identity or appearance of a subject is preserved between input and output images.
>
> In contrast, our proposed TI2I task aims to **bridge the gap between visual understanding and generation**, rather than enforce pixel-level similarity. Given a sample triplet (T, $I_1$, $I_2$), the model is trained to **first understand both the text T and the input image $I_1$**, and then generate a new image $I_2$ conditioned on both modalities. Pixel-level consistency is **not required**-only **semantic alignment**.
>
> For example, if $I_1$ is “a picture of grass,” $I_2$ is “a dog running on grass,” and the text is “A dog running on \<image\>,” the model is expected to understand that “\<image\>” refers to “grass” and generate an appropriate scene. The grass in $I_1$ and $I_2$ may differ visually, as long as they share the same **semantic concept**. This formulation allows the model to **learn multimodal alignment and reasoning** within a single training sample, and gradually unifies image understanding and generation in a **semantically coherent** way.
>
> We also give some **prompt format design and application in Appendix Figure 1**, where the model trained on the TI2I task can be adapted to subject-driven generation and zero-shot style transfer.
>
> **Q3: Several images in Figure 7 are not well aligned with the prompt.**
>
> Ans: Thanks for pointing this out. In Figure 7, we used a set of prompts collected from the web and prior works, and generated the images without cherry-picking. While many results are reasonable, some generations are not well aligned with the prompts. This is primarily due to limited training data and compute budget. Our current model is not strong enough to handle some unseen prompts, and may produce incorrect results. We believe it can be mitigated by scaling up training data and incorporating more diverse scenarios in the future work.
>
> **Q4: Why AR-DTok shows better capability in generating complete and correct text within images in Figure 6?**
>
> Ans: We believe this phenomenon is due to two key factors:
> **(a) Training data.** As shown in Appendix Table 1, AR-DTok is trained with more data than Dif-DTok (50+23M vs 23M). This provides AR-DTok with **better alignment** to TA-Tok tokens, particularly for fine-grained content such as text within images.
>
> **(b) Architecture difference.** As shown in Figure 4, the two de-tokenizers differ in how they integrate TA-Tok tokens:
> - In AR-DTok, TA-Tok tokens z and VQVAE tokens y are concatenated as [y, z] and processed jointly via **self-attention** in an autoregressive manner.
> - In Dif-DTok, TA-Tok tokens are injected  through **cross-attention** with the noise input during denoising.
>
> Because TA-Tok tokens carry **explicit** 2D structure and visual details, the **self-attention fusion** in AR-DTok is more effective in learning fine-grained **token correspondences** between TA-Tok and VQVAE. This contributes to AR-DTok’s superior ability in generating coherent and correct text within images.
>
> **Q5: Evaluation of SAP design for I2I and TI2I tasks.**
>
> Ans: As discussed in Q2, the proposed I2I and TI2I tasks are designed as **auxiliary tasks** to mitigate the gap between visual understanding and generation, and are different from traditional TI2I tasks. Therefore, we evaluate the design of SAP on image editing, a traditional TI2I task. Specifically, we simply finetune Tar 1.5B on OmniEdit [1] and evaluate it on Emu-Edit [2]. The results are shown below.
> We find that different SAP settings perform **similarly on CLIP-I and CLIP-T**, which measures the semantic similarity between the input image, prompt and edited image. However, for the L1 metric, which reflects pixel-level consistency, using **more tokens** (e.g., 729) provides better reconstructions. We will include the evaluation of TI2I tasks in the revision.
>
> | #Token | CLIP-I↑ | CLIP-T↑ | L1↓ |
> |---|---|---|---|
> | 729 | 0.839 | 0.279 | 0.131 |
> | 169 | 0.841 | 0.279 | 0.160 |
> | 81 | 0.845 | 0.278 | 0.224 |
>
>
> [1] Wei, Cong, et al. Omniedit: Building image editing generalist models through specialist supervision. ICLR.
>
> [2] Sheynin, Shelly, et al. Emu edit: Precise image editing via recognition and generation tasks. CVPR.

---

> > ### Author Response · Authors · 2025-08-07
> > **Gentle Reminder**
> >
> > Dear Reviewer GJa8,
> >
> > We sincerely appreciate the time and effort you have dedicated to reviewing our paper. In response to your comments, we have provided detailed clarifications and conducted additional experiments to address your specific concerns.
> >
> > As the discussion period will conclude in a few days, we would be truly grateful if you could kindly take a moment to review our rebuttal. If you find that our responses have resolved the issues you raised, we would sincerely appreciate it if you could consider raising the score.
> >
> > Thank you once again for your thoughtful feedback and valuable contribution to our work.
> >
> > Best regards,
> >
> > The Authors

---

> > ### Comment · Reviewer_GJa8 · 2025-08-07
> >
> > I appreciate the authors' response and complementary experiments. As detailed in the "final justification", my final score falls between "borderline accept" and "accept", and I raise score to "accept" on the condition that the authors adequately incorporate the discussed points and complementary experiments into the final version of the manuscript.

---

> > > ### Author Response · Authors · 2025-08-07
> > > **Thank You for the Score Adjustment. We Will Revise Accordingly**
> > >
> > > Dear Reviewer GJa8,
> > >
> > > Thank you very much for your constructive feedback and for conditionally raising your score to Accept. We truly appreciate your recognition and support.
> > >
> > > We will carefully revise the final version paper to address all the concerns raised during the review process. Specifically, we will clarify the performance gap in Table 6, better distinguish our TI→I task from traditional formulations, address the mismatches in Figure 7, provide further insights into AR-DTok’s behavior, and include additional evaluations of the SAP design.
> > >
> > > We truly appreciate your constructive suggestions and support.
> > >
> > > Best regards,
> > >
> > > The Authors

---

### Official Review · Reviewer_xYB2 · 2025-07-02

**Clarity:** 3
**Significance:** 3
**Originality:** 3
**Rating:** 4
**Confidence:** 3

**Summary:**

This paper introduces a novel multimodal framework designed to unify visual understanding and generation through discrete representations. Specifically, the authors leverage a pre-trained LLM to generate an initial codebook, selecting representative codes based on their average distance from others. Subsequently, the framework aligns visual and textual feature spaces by training a SigLIP2 model alongside a projector. To capture multi-scale visual features, a scaling-adaptive strategy is adopted, and the frozen SigLIP2 model serves as a teacher to provide reconstruction targets, jointly guiding the model’s optimization. For generation tasks, the framework integrates two mainstream generative models—autoregressive and diffusion models—externally to enable image generation.

**Questions:**

1. Figure 5 shows that Janus Pro performs considerably worse in generation quality under the same data scale. What factors contribute to this discrepancy?

2. Given that visual information is generally considered more complex than textual information, why is the visual codebook smaller than its textual counterpart? Furthermore, considering the semantic granularity differences between modalities, is it appropriate to assume a shared visual-textual feature space?

3. Based on the generation results, Tar seems to exhibit weak identity consistency. What might be causing this issue?

**Ethical Concerns:**

["NO or VERY MINOR ethics concerns only"]

**Final Justification:**

After carefully reviewing the comments from fellow reviewers and the authors' response, I believe this paper is above the acceptance threshold for NeurIPS. Therefore, I will maintain my initial rating.

**Limitations:**

yes

**Quality:**

3

**Strengths And Weaknesses:**

Strong:

1. The paper proposes an innovative multimodal framework that unifies visual understanding and generation based on discrete tokens. Its approach of constructing a visual codebook from the LLM's feature space may provide meaningful insights for future work in this domain.

2. The proposed method, Tar, demonstrates performance close to SOTA levels across several benchmarks and significantly outperforms Janus Pro on generative metrics.

Weakness:

1. Lines 120–121 mention that representative embeddings are selected based on their average distance to others. Is this achieved through clustering?

2. The paper states that 150k vocabularies are excessively large, yet later in the appendix, it is mentioned that the model uses 65k vocabularies, an amount not substantially smaller. This appears contradictory.

3. Additionally, the ablation study does not report results for the configuration in which random initialization is expanded to 65,536 tokens. Why is this omitted?

4. A further concern arises from the comparison between the first and third configurations: performance from random initialization and LLM-based initialization appears similar. How should this be interpreted?

---

> ### Author Rebuttal · Authors · 2025-07-31
>
> Dear Reviewer xYB2,
>
> Thank you very much for your helpful comment!
>
> **Q1: Are representative LLM embeddings selected through clustering?**
>
> Ans: We do not use clustering algorithms (e.g., K-Means) for selection. Instead, we adopt a simple and efficient **distance-based ranking** strategy. Specifically, we compute the average cosine similarity between each token embedding and all others, and select the tokens with the lowest average similarity (i.e., most representative and diverse). The selection process is shown below and will be included in the revision:
>
> ```
> # embeddings.shape=[150K, D]
> norms = torch.norm(embeddings, dim=1, keepdim=True)
> cosine_sim = torch.matmul(embeddings, embeddings.T) / (norms * norms.T)
> average_similarity = torch.mean(cosine_sim, dim=1)
> topk=65536
> representative_tokens = torch.argsort(average_similarity)[:topk]
> ```
>
> This method allows us to efficiently obtain a **diverse and informative subset** of the LLM's embedding space, forming the initial visual vocabulary.
>
> **Q2: The 65K vocabulary is still large.**
>
> Ans: We agree that 65K may still seem large. However, in vector quantization, **a larger vocabulary is generally beneficial**, which helps **preserve more visual detail** and **reduce quantization error**. Ideally, we would use the full 150K vocabulary from the LLM.
>
> In practice, though, using all 150K high-dimensional embeddings (≥1536) results in **excessive GPU memory usage** during TA-Tok training. To address this, we reduce the vocabulary size to 65K, so that a 40GB A100 GPU can hold it and keep training stable.
>
> **Q3: Where is the result of randomly initialized codebook expanded to 65536 tokens?**
>
> Ans: Traditional VQ codebooks typically use smaller vocabularies (8K–64K) with low-dimensional embeddings (e.g., 4–32). In contrast, to preserve the semantic richness of SigLIP features (1152-D), we increase the codebook dimension to 1536.
> However, this combination of **large vocabulary size and high dimension** is **extremely difficult to train** when initialized randomly. As shown in Appendix Table 3, even with 32,768 tokens and 1536-D (setting (b)), the model **struggles to converge** due to codebook collapse. Expanding the random-initialized codebook to 65,536 tokens only **worsens this issue**, making training **infeasible** in our experiments.
> This challenge further motivates our use of LLM-initialized embeddings, which offer much more stable and meaningful training dynamics.
>
>
> **Q4: Performance from random initialization and LLM-based initialization appears similar.**
>
> Ans: First, the improvement of LLM-based initialization compared to random initialization is still significant after sufficient training, about 2.7%, 13% improvement on the challenging GQA and MME respectively, which are **non-trivial** for these tasks.
>
> More importantly, under **data-limited settings**, the benefit becomes more significant. As shown in the table below, when we reduce the training data from 25M to 12.5M and 6.25M samples, **LLM-initialized codebook consistently outperforms random initialization**, especially on tasks that rely more heavily on semantic alignment, such as MME and GQA. This highlights that LLM-based initialization helps **align TA-Tok with the LLM’s embedding space more efficiently**, particularly in low-resources scenarios.
>
> | Data | setting | GQA | MME | POPE | DPG |
> |---|---|---|---|---|---|
> | 12.5M | random | 53.5 | 965 | 85.8 | 66.0 |
> |  | LLM | 57.2(+6.9%) | 1258(+30%) | 86.8(+1.2%) | 67.7(+2.6%) |
> | 6.25M | random | 49.1 | 868 | 84.5 | 60.6 |
> |  | LLM | 55.5(+13%) | 1235(+42%) | 86.3(+2.1%) | 62.0(+2.3%) |
>
> **Q5: Why Janus performs worse in generation quality in Figure 5?**
>
> Ans: As noted in Line 246-247, the ‘Janus’ baseline used here is **different from the original Janus Pro** [6], it denotes a baseline method which adopts SigLIP for image understanding and VQVAE for image generation.
>
> As illustrated in Line 256-257, we believe the poor generation performance stems from a **conflict** between the two visual representations: SigLIP (semantic) and VQVAE (pixel-level). Since these representations are **not aligned**, the training signals from image understanding and generation tasks **do not reinforce each other**.
>
> This hypothesis is further supported in **Table 6**, where **joint training fails to improve** Janus on either task, while it brings **significant gains** for both VQVAE baseline and our proposed method. This highlights the benefit of using a unified visual representation, as enabled by our TA-Tok design.
>
> **Q6: Why is the visual codebook smaller than the text codebook?**
>
> Ans: As mentioned in Q2, the visual codebook size reflects a **trade-off between encoding quality and computational cost**. While a large codebook can better capture visual details, it significantly increases the **computational burden** during both tokenizer and MLLM training.
> Moreover, increasing the codebook size will also encounter the low codebook usage issue.
>
> **Q7: Is it appropriate to assume a shared visual-textual feature space?**
>
> Ans: Thanks for your thoughtful question! We agree that visual and textual modalities differ in semantic granularity. Ideally, a unified model should capture both pixel-level detail and semantic-level features.
>
> However, in practice, integrating pixel-level information directly into fully semantic LLMs is **extremely resource-intensive and technically challenging**. As shown in Figure 5, methods like VQVAE, which focus on pixel-level reconstruction, fall behind Hybrid approaches and our method in both understanding and generation tasks.
>
> To **achieve better cross-modal alignment**, we use a **shared semantic feature space**, where visual and textual tokens are **compatible**. We then use a lightweight de-tokenizer to project these semantic visual tokens back to the pixel space for image generation. This setup allows us to **maintain semantic consistency** while still supporting high-quality image synthesis.
>
> **Q8: Tar’s generation results are weak in identity consistency?**
>
> Ans: Thank you for the observation. As discussed in our Limitations (Appendix Sec. I), autoregressive and diffusion models are **not original designed for image reconstruction**, but in our framework, we repurpose them as de-tokenizers for this task. Since they are not explicitly optimized for preserving fine-grained identity features, some loss in identity consistency may occur.
>
> To address this, future work can explore **adding local consistency constraints**, such as token-to-token or patch-to-path supervision,which may better preserve identity than the current image-level objectives.

---

> > ### Author Response · Authors · 2025-08-07
> > **Have We Sufficiently Addressed Your Concerns?**
> >
> > Dear Reviewer xYB2,
> >
> > Thank you very much for your detailed and constructive review. We sincerely appreciate the time and thought you put into evaluating our work.
> >
> > We are grateful for your Borderline accept recommendation and the thoughtful feedback you provided. We have carefully addressed all the concerns you raised, including clarifications regarding the embedding selection strategy, vocabulary size choices, ablation design, initialization comparisons, generation performance, and identity consistency.
> >
> > While we certainly **do not wish to take more of your time than necessary**, your engagement is **extremely valuable to the final assessment** of our work. If the responses we’ve provided have resolved your concerns, we would deeply appreciate it if you would consider raising your score. Of course, if there are any remaining questions or issues, we would be more than happy to continue the discussion.
> >
> > Thank you again for your contribution to the review process.
> >
> > Best regards,
> >
> > The Authors

---

> > > ### Comment · Reviewer_xYB2 · 2025-08-08
> > >
> > > Thank you to the authors for their detailed and thoughtful response. Most of my concerns have been satisfactorily addressed. A few minor issues remain—for instance, the distance-based ranking strategy appears to be a simplified variant of DPC clustering, and it may be worthwhile to consider replacing it with a more robust clustering algorithm. Nonetheless, I am overall satisfied with the quality of the paper and the authors' response, and I will maintain my current rating.

---

> > > > ### Author Response · Authors · 2025-08-08
> > > > **Thank You for Your Recognition and Feedback!**
> > > >
> > > > Dear Reviewer xYB2,
> > > >
> > > > Thank you for your feedback and for acknowledging that most of your concerns have been satisfactorily addressed.
> > > >
> > > > We agree with your point regarding the distance-based ranking strategy. Due to limited time during the rebuttal phase, we could not fully explore this aspect, but we will carefully consider and study more robust clustering methods in the revision.
> > > >
> > > > Thank you again for your valuable input and the time you have dedicated to reviewing our work.
> > > >
> > > > Best regards,
> > > >
> > > > The Authors

---

### Official Review · Reviewer_fcrb · 2025-07-07

**Clarity:** 3
**Significance:** 2
**Originality:** 3
**Rating:** 4
**Confidence:** 5

**Summary:**

This paper tries to unify the visual understanding and generation within a shared discrete semantic representation. The core design is the Text-Aligned Tokenizer, which builds a discrete semantic visual representation initialized from LLM embeddings. This paper also proposes the Scale-Adaptive Pooling to adjust the number of tokens for visual understanding and generation. Finally, autoregressive and diffusion Generative De-Tokenizers are introduced to enable visual generation.

**Questions:**

Please refer to the Weakness I have outlined and provide clarifications for each point individually. I would love to raise the rating is my concerns are well-addressed.

**Ethical Concerns:**

["NO or VERY MINOR ethics concerns only"]

**Final Justification:**

Although I’m not fully convinced, I think it is a meaningful exploration backed by solid experimental support. I choose to raise the score.

**Limitations:**

Yes.

**Paper Formatting Concerns:**

No.

**Quality:**

3

**Strengths And Weaknesses:**

[Strength]

1. The thinking of the visual representation outlined in the introduction is clear and insightful.

2. The paper is well-written and easy to understand. The experiments and ablation studies are detailed and comprehensive.

[Weakness]
1. **My biggest concern is the motivation and rationale behind the proposed TA tokenizer**: (1) The continuous SigLIP2 features already provide rich semantic information. As shown in Section 4.3 and Figure 5, the proposed visual representations actually lead to worse visual understanding performance. (2) If the authors’ goal is to use a single visual representation, it is unclear why this representation must be discrete. Methods like MetaMorph [1] and MetaQuery [2] use continuous single visual representations effectively. This paper lacks detailed comparison with them in Section 4.3. Methods like VILA-U [3] and UniTok [4] employ discrete representations because they leverage unified tokenizers, where the generated tokens can be directly de-tokenized into images. However, in this paper, the generated tokens are only used as conditions, and the so-called de-tokenizer still needs to synthesize the image based on these conditions. This undermines the justification for requiring discrete tokens.

2. The authors emphasize in line 32 that "both modalities are learned within a single latent space." However, I question the validity of this statement. In this paper, the visual generation process is not end-to-end: one model is used to generate the condition, and a separate model is used to project that condition into images.

3. The *Understanding Only Model* listed in Table 9 are outdated, and comparisons with them are not meaningful. The authors should include state-of-the-art models for a more relevant and fair evaluation.

4. In lines 163–169, the authors compare the training and inference efficiency of autoregressive and diffusion models, but provide no supporting evidence. I am not convinced by this conclusion. To strengthen the argument, the authors should include empirical data such as training time and inference latency.

[1] Tong, Shengbang, et al. "Metamorph: Multimodal understanding and generation via instruction tuning." arXiv preprint arXiv:2412.14164 (2024).

[2] Pan, Xichen, et al. "Transfer between modalities with metaqueries." arXiv preprint arXiv:2504.06256 (2025).

[3] Wu, Yecheng, et al. "Vila-u: a unified foundation model integrating visual understanding and generation." arXiv preprint arXiv:2409.04429 (2024).

[4] Ma, Chuofan, et al. "Unitok: A unified tokenizer for visual generation and understanding." arXiv preprint arXiv:2502.20321 (2025).

---

> ### Author Rebuttal · Authors · 2025-07-31
>
> Dear Reviewer fcrb,
>
> Thank you very much for your thoughtful and constructive review.
>
> **Q1: Comparisons to continuous visual representations (MetaMorph and MetaQuery)**
>
> Ans: We appreciate your suggestion to clarify the motivation behind using discrete representations, and we will expand in the revision to include a more detailed  comparison with MetaMorph and MetaQuery.
>
> **(1) MetaMorph** builds a visual representation by regressing SigLIP latents through an additional vision head, followed by training a diffusion decoder with an image autoencoding objective. Due to this **regression and reconstruction pipeline**, MetaMorph is not able to **sample diverse images** from a prompt. It lacks a natural generative process like that in autoregressive LLMs. In contrast, our discrete visual tokens allow **direct integration with autoregressive sampling strategies**, enabling diverse and controllable generation in a unified token space.
>
> **(2) MetaQuery** leverages a frozen MLLM for multimodal condition extraction. However, the image understanding and generation modules remain **decoupled**. The former is handled by the frozen MLLM, the latter by a diffusion decoder. This is **conceptually closer to traditional diffusion-based methods** (e.g., DiT) than to a unified model with shared representations. In contrast, our method explicitly aims to unify understanding and generation through a single discrete semantic space (i.e., TA-Tok).
>
> **(3)** Why do we refer to Dif-DTok as a “De-Tokenizer” instead of a diffusion decoder (generator)?
> Our Dif-DTok is trained to **reconstruct** the input image from discrete visual tokens produced by TA-Tok, rather than to generate new images conditioned on TA-Tok’s tokens. Since TA-Tok tokens already **preserve 2D structure and rich visual details** (e.g., 27*27=729 tokens 2D grid), Dif-DTok’s role is to **faithfully** transform these structured tokens into the pixel space. We believe “de-tokenizer” more accurately captures its function as part of a modular pipeline, **rather than a full generative decoder**.
>
> **(4) The generated visual tokens are not merely conditioning signals, but 2D visual tokens that encode the full information necessary for image reconstruction—including structure and detail.**
> This is demonstrated in Appendix Table 5, where images encoded by TA-Tok can be **faithfully reconstructed by a De-Tokenizer**. We believe it is a **key difference** from MetaQuery. Our Tar model can **directly compose and organize visual content** in discrete token space, similar to how LLMs generate coherent text sequences. Importantly, our system **does not rely on a strong diffusion model** to handle both multimodal prompt understanding and image generation like MetaQuery. Instead, a simple De-Tokenizer that reliably converts TA-Tok’s tokens into VQ-VAE or VAE tokens is sufficient.
>
> Therefore, we consider our method to be **in line with prior work such as VILA-U and UniTok**, but with a more expressive and **LLM-aligned tokenizer design**. While we acknowledge that discrete tokens currently underperform continuous features in some understanding tasks. However, as noted in Appendix I, we see this as a **technical rather than conceptual limitation**. We expect the gap to narrow with longer token sequences, higher image resolutions, and improved vector quantization methods.
>
> **Q2: Motivation to use discrete tokens?**
>
> Ans: The community recognizes the value of developing a unified model for both visual understanding (VLM) and generation. Among potential architectures, **LLMs provide the most natural and scalable foundation** for unifying these tasks-offering a flexible framework centered on **next-token prediction**.
>
> The central challenge is how to **represent visual information within the LLM paradigm**. While approaches like Transfusion adopt separate diffusion objectives for generation, they break the unified modeling pipeline for visual understanding and generation. In contrast, **discretizing images into tokens enables the use of a single autoregressive modeling strategy, next-token prediction, for both text and image modalities.**
>
> However, existing discrete visual tokenizers (e.g., VQVAE) tend to operate at the pixel level, which **limits their semantic alignment with LLMs**. Unified tokenizers like VILA-U and UniTok attempt to achieve two **fundamentally conflicting objectives simultaneously**: pixel-level reconstruction and semantic-level alignment, making it difficult to strike a good balance between the two. To address this, we propose TA-Tok, a tokenizer designed to produce **fully semantic, discrete** visual tokens, initialized from LLM embedding. This alignment allows visual tokens to **seamlessly interact** with language tokens during autoregressive modeling.
>
> While TA-Tok does not directly decode images, it preserves **rich 2D spatial and structural information**. This makes the output tokens **more than just abstract conditions**: they encode image layout and visual detail in a structured form. Then a lightweight de-tokenizer suffices to reconstruct the image, unlike previous works that require powerful diffusion decoders to synthesize images from scratch.
>
> In summary, **discrete tokens are essential** for unifying visual understanding and generation within an LLM framework. Our design of TA-Tok bridges the semantic gap between vision and language, enabling a more integrated and efficient modeling pipeline.
>
> **Q3: Whether both modalities are learned within a single latent space?**
>
> Ans: In Line 32, we emphasize the advantages of a unified visual representation for both image understanding and generation, when compared to approaches that rely on **two types of visual tokens** (e.g., Janus). Our approach aims to unify modalities at the **token level**, where both vision and language operate in a compatible discrete semantic space.
>
> As clarified in our response to Q1 and Q2, the de-tokenizer in our pipeline **does not serve as a standard generative model**, but rather transforms the semantic tokens from TA-Tok into VQVAE/VAE latent space for image reconstruction. The **planning of image structure and visual details**-traditionally done by the image generator-is instead **handled by the LLM** through discrete token generation.
>
> Methods like VILA-U and UniTok also claim to unify modalities with simple CNN decoders. In contrast, we introduce autoregressive and diffusion-based de-tokenizers, which are more expressive and better suited to decoding semantic visual token from TA-Tok. Thus, despite architectural differences, our method **belongs to the same line** of unified modeling works like VILA-U and UniTok, but pushes further in terms of semantic alignment and generation fidelity.
>
> **Q4: The Understanding Only Models in Tab.1 are outdated.**
>
> Ans: We followed prior works [6, 55] in selecting these baselines and will include more recent models in the revision. However, Table 1 mainly aims to **compare unified models**, with understanding-only models serving as **reference points**. Direct comparison is also challenging, as recent understanding models often use more data and complex training pipelines. Despite this, our Tar model shows **state-of-the-art performance among unified approaches**.
>
> **Q5: Empirical results of training and inference latency of two De-Tokenizers.**
>
> Ans: Thank you for pointing this out. We have conducted a comparison between AR-DTok and Dif-DTok in both training and inference phases, as summarized in the table below. While the two models differ in architecture, parameters, and codebases, we carefully controlled for factors such as batch size, token count, and hardware setup to ensure a fair comparison. Some key observations:
> - Training speed is comparable across both models.
> - Inference: AR-DTok uses less GPU memory and runs faster at 256 tokens, but becomes slower at 1024 tokens due to its autoregressive nature.
> - Dif-DTok scales better at high token counts but consumes more memory.
>
> We will include this analysis and clarify the differences in the revision.
>
> | Method | Param | #Token | Train Speed (b/s) | Inference Speed (s/b) | Inference Memory |
> |---|---|---|---|---|---|
> | AR-DTok | 775M | 256 | 2.85 | 3.45 | 26.1GB |
> | Dif-DTok | 600M | 256 | 2.92| 3.63 | 34.6GB |
> | AR-DTok | 775M | 1024 | 1.84 | 14.14 | 31.2GB |
> | Dif-DTok | 600M | 1024 | 1.90 | 9.77 | 44.9GB |
>
> b/s: batch per second. s/b: second per batch. The batch size is 6 for training and 4 for inference.

---

> > ### Author Response · Authors · 2025-08-07
> > **Gentle Reminder**
> >
> > Dear Reviewer fcrb,
> >
> > We sincerely appreciate the time and effort you have dedicated to reviewing our paper. In response to your comments, we have provided detailed clarifications and conducted supporting experiments to address your specific concerns.
> >
> > As the discussion period will conclude in a few days, we would greatly appreciate it if you could kindly take a moment to review our rebuttal at your convenience. If there are any remaining questions or suggestions, we are more than happy to address them promptly.
> >
> > Thank you once again for your thoughtful feedback and valuable contribution to our work.
> >
> > Best regards,
> >
> > The Authors

---

> ### Comment · Reviewer_fcrb · 2025-08-08
>
> Thank you to the authors for their detailed rebuttal. They shared additional insights on their design which I am greatly appreciated.  Maybe the "De-Tokenizer" occupies a middle ground, functioning in between as a generation model and a reconstruction model. While I’m not fully convinced, I think it is a meaningful exploration backed by solid experimental support. As a result, I have decided to raise my rating to borderline accept.

---

### Decision · Program_Chairs · 2025-09-17

**Decision:**

Accept (poster)

**Comment:**

This work aims to unify visual understanding and generation via discrete vision tokens. To align with the vision and text space, vision tokens are initialized by projecting from the discrete text tokens. Meanwhile, the vision tokens are employed to generate images by AR-DTok/Dif-DTok. Although the initial review scores are mixed, the rebuttal addressed many concerns. All reviewers find that the exploration in this work is novel and the empirical results are solid. Therefore, all reviewers have positive scores after rebuttal. Please incorporate the comments and suggestions from reviewers into the final version and improve the presentation accordingly.